# HESS Opinions: Towards a common vision for the future of hydrological observatories

**Paolo Nasta**[1], **Günter Blöschl**[2,17], **Heye R. Bogena**[3], **Steffen Zacharias**[4], **Roland Baatz**[5], **Gabriëlle De Lannoy**[6], **Karsten H. Jensen**[7], **Salvatore Manfreda**[8], **Laurent Pfister**[9,10], **Ana M. Tarquis**[11], **Ilja van Meerveld**[12], **Marc Voltz**[13], **Yijian Zeng**[14], **William Kustas**[15], **Xin Li**[16], **Harry Vereecken**[3], and **Nunzio Romano**[1]

[1]Department of Agricultural Sciences, Division of Agricultural, Forest and Biosystems Engineering,
University of Naples Federico II, Portici (Naples), Italy

[2]Institute of Hydraulic Engineering and Water Resources Management, TU Wien, Vienna, Austria

[3]Agrosphere Institute, Forschungszentrum Jülich GmbH, Jülich, Germany

[4]Helmholtz Centre for Environmental Research – UFZ, Leipzig, Germany

[5]Leibniz Centre for Agricultural Landscape Research (ZALF), Müncheberg, Germany

[6]Department of Earth and Environmental Sciences, Division Soil and Water Management, KULeuven, Leuven, Belgium

[7]Department of Geosciences and Natural Resource Management, Geology section,
University of Copenhagen, Copenhagen, Denmark

[8]Department of Civil, Building and Environmental Engineering (DICEA), University of Naples Federico II, Naples, Italy

[9]Luxembourg Institute of Science & Technology (LIST), Environmental Sensing and Modelling unit (ENVISION),
Esch-sur-Alzette, Luxembourg

[10]Department of Engineering, Faculty of Science, Technology and Medicine (FSTM), University of Luxembourg,
Luxembourg, Luxembourg

[11]CEIGRAM, Department of Applied Mathematics, Universidad Politécnica de Madrid (UPM), Madrid, Spain

[12]Department of Geography, University of Zurich, Zurich, Switzerland

[13]Laboratoire sur les Interactions Sol-Agrosystème-Hydrosystème
UMR INRAE-IRD-Institut Agro 2, Montpellier CEDEX, France

[14]Department of Water Resources, Faculty of Geo-Information Science and Earth Observation (ITC), University of Twente,
Enschede, the Netherlands

[15]USDA-ARS Hydrology & Remote Sensing Lab, Beltsville, USA

[16]National Tibetan Plateau Data Center, State Key Laboratory of Tibetan Plateau Earth System,
Environment and Resources, Institute of Tibetan Plateau Research, Chinese Academy of Sciences, Beijing 100101, China

[17]Department of Civil, Chemical, Environmental and Materials Engineering, University of Bologna, Bologna, Italy

**Correspondence:** Nunzio Romano (nunzio.romano@unina.it)

**Abstract.** The Unsolved Problems in Hydrology (UPH) initiative has emphasized the need to establish networks of multi-decadal hydrological observatories to gain a deep understanding of the complex hydrologic processes occurring within diverse environmental conditions. The already existing monitoring infrastructures have provided an enormous amount of hydrometeorological data, facilitating detailed insights into the causal mechanisms of hydrological processes, the testing of scientific theories and hypotheses, and the revelation of the physical laws governing catchment behavior. Yet, hydrological monitoring programs have often produced limited outcomes due to the intermittent availability of financial resources and the substantial efforts required to operate observatories and conduct comparative studies to advance previous findings. Recently, some initiatives have emerged that aim to coordinate data acquisition and hypothesis test-

ing to facilitate an efficient cross-site synthesis of findings. To this end, a common vision and practical data management solutions need to be developed. This opinion paper provocatively discusses two potential endmembers of a future hydrological observatory (HO) network based on a given hypothesized community budget: a comprehensive set of moderately instrumented observatories or, alternatively, a small number of highly instrumented supersites.

A network of moderately instrumented monitoring sites would provide a broad spatial coverage across the major pedoclimatic regions by supporting cross-site synthesis of the lumped hydrological response (e.g., rainfall–runoff relationship, Budyko analysis) across diverse continental landscapes. However, the moderate instrumentation at each site may hamper an in-depth understanding of complex hydrological processes. In contrast, a small number of extensively instrumented research sites would enable community-based experiments in an unprecedented manner, thereby facilitating a deeper understanding of complex, non-linear processes modulated by scale-dependent feedback and multiscale spatiotemporal heterogeneity. Lumping resources has proven to be an effective strategy in other geosciences, e.g., research vessels in oceanography and drilling programs in geology. On the downside, a potential limitation of this approach is that a few catchments will not be representative of all pedoclimatic regions, necessitating the consideration of generalization issues.

A discussion on the relative merits and limitations of these two visions regarding HOs is presented to build consensus on the optimal path for the hydrological community to address the UPH in the coming decades. A final synthesis proposes the potential for integrating the two endmembers into a flexible management strategy.

Keywords: hydrological observatory network, experimental catchments, cross-site synthesis, hypothesis testing vs. exploratory science, unsolved problems in hydrology, societal needs, technology advancements.

---

*Highlights.*

- The historical situation of hydrological observatories (HOs) has led to fragmented knowledge and sub-optimal research progress.

- Some initiatives have emerged to coordinate and standardize data and models, resulting in efficient cross-site synthesis.

- It is important to stimulate discussion within the hydrological community to arrive at a consensus view on HOs.

**Graphical abstract**

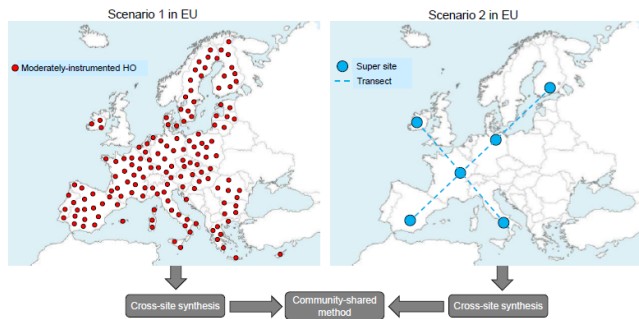

## 1   How do we advance scientific understanding of hydrological processes?

[TS1] Water is under increasing threat due to human activities. Rapid changes in land use, such as the adoption of more intensive farming practices, the expansion of urbanization, and the abandonment of land in rural areas, have a significant impact on the hydrological cycle and water quality, whereas unsustainable water withdrawals lead to the depletion of resources. Global warming is expected to exacerbate hydrological extremes, resulting in more disastrous floods and severe droughts that will further threaten water security. In light of these challenges, the mission of hydrologists and water managers is to sustainably meet human needs while preserving biodiversity and ecosystem services based on the most accurate and up-to-date information. However, the extent to which anthropogenic stressors influence the hydrologic cycle is not yet fully understood, and the effectiveness of adaptation actions to guide the management of water resources has yet to be fully evaluated. Hydrology is a data-hungry discipline, but the limited observations on all components of the terrestrial hydrosphere, from bedrock to the lower atmosphere, represent a significant obstacle to progress in the understanding of hydrologic process dynamics.

To grasp the daunting complexity of the hydrological cycle, particularly in relation to the impact of human activities on the critical zone and catchment functionality, and to address the Unsolved Problems in Hydrology (UPH), several hydrological observatories (HOs) have been established around the world with the specific purpose of monitoring hydrological states and flows (Blöschl et al., 2019; Arora et al., 2023).

The concept of hydrological observatories (HOs) can be traced back to the early 1900s when scientists began to recognize the significance of long-term data collection for understanding hydrological processes (McDonnell et al., 2007). In 1903, runoff and other hydrological variables were initially collected in the Sperbelgraben and Rappengraben experimental catchments in the Emmental region of Switzerland. These catchments remain operational and hold one of the longest continuous discharge records in the world

(Stähli et al., 2011). In the United States, the first HOs were the Wagon Wheel Gap Experiment in Colorado (Bates and Henry, 1928), the Coweeta Hydrological Laboratory in North Carolina (Neary et al., 2012), and a catchment network across the continental US established by the USDA Agricultural Research Service (Goodrich et al., 2021). These sites were designed to study the influence of human activities on hydrological systems, with a particular focus on deforestation and afforestation, land use changes, and agricultural practices (Whitehead and Robinson, 1993). Since the 1950s, there has been a notable increase in the number of HOs established across the globe. The HO sites have provided invaluable information for the effective management of water resources. Currently, a diverse range of entities, including government agencies (e.g., the Hydrologic Benchmark Network of the US Geological Survey), universities and research institutions, international organizations, and non-governmental organizations, provide funding and support for these sites.

## 2   Building integrated observation platforms

A considerable number of rivers worldwide have been equipped with gauges for governmental agencies to monitor precipitation and streamflow for the purpose of water management. The data collected have been primarily utilized at the national level, although there are several transnational initiatives, including the Global Runoff Data Centre in Koblenz, Germany, and the Camels datasets, such as those for the US, Chile, and Brazil, (Addor et al., 2017; Alvarez-Garreton et al., 2018; Chagas et al., 2020). HOs extend beyond these conventional networks, striving to gain a more comprehensive understanding of hydrological processes, typically in smaller catchments.

A hydrological observatory is defined as a cyber-physical infrastructure established within a catchment area to monitor the hydrological variables and fluxes, as well as to characterize the hydrological behavior of the three-dimensional spatial domain. The catchment is assumed to be the fundamental hydrological unit, with well-defined system boundaries. It is from this unit that the impact of anthropogenic disturbances (global warming, land use change, aquifer contamination, etc.) on water resources can be evaluated through a long-term data analysis. Given the impracticality of full catchment coverage, the hydrological observatory focuses on a selected cluster of sub-catchments (spatial resolution of hectares) which are representative of land use, geomorphology, topography, and pedology similarities (Bogena et al., 2006). Consequently, the selected sub-catchments are equipped with wireless sensor networks for continuous data collection and are subjected to disparate field campaigns, contingent upon budgetary constraints.

The selection of sensors is crucial for the effective collection of hydrometeorological data within a hydrological observatory. Weather station networks (also called synoptic stations) ensure the collection of meteorological data and have been integrated in many countries with weather radar networks for the purpose of detailed precipitation estimation (Sokol et al., 2021). Snow water equivalent is already measured on a routine basis with snow pillows (e.g., by the SNOTEL network in the United States) or, more experimentally, by airborne lidar snow depth surveys (Painter et al., 2016). Groundwater levels are monitored on a routine basis, whereas distributed temperature sensing technology is a more novel approach for estimating infiltration rates and, potentially, catchment-scale groundwater recharge (Medina et al., 2020). The measurements of soil water content and matric potential, soil temperature, and soil bulk electrical conductivity are conducted across soil profiles at the point scale (Hoffmann et al., 2015; Peng et al., 2019; Bogena et al., 2022). Cosmic-ray neutron sensors, meanwhile, are capable of extending the footprint of soil moisture to approximately 150–200 m in radius (Romano, 2014; Köhli et al., 2015; Baatz et al., 2017). At experimental sites, surface and subsurface runoffs from hillslopes are measured using flowmeters in runoff plots (Fu et al., 2024). In addition, the mapping of saturation areas on hillslopes (Silasari et al. 2017) and channel–network dynamics (Jensen et al., 2019; Strelnikova et al., 2023; Noto et al., 2024) provide insight into the spatial patterns of catchment-scale processes that extend beyond point measurements. Topographic surveys assist in determining surface flow paths within a catchment, thus enabling the extension of point measurements to the catchment scale (e.g., Rinderer et al., 2019; Fan et al., 2019; Refsgaard et al., 2021). The rates of soil erosion and deposition are quantified through the use of sediment fences, soil profile surveys, and cosmogenic nuclide analysis, in addition to repeated high-precision topographic surveys. The measurement of soil physical, chemical, and hydraulic properties is typically conducted in field campaigns and laboratory experiments, with remote sensing serving as a complementary technique. Geophysical tools, such as electromagnetic (EM) surveys, have the potential to provide valuable insights into the imaging of aquifer systems and the characterization of subsurface heterogeneity (Nasta et al., 2019; Dewar and Knight, 2020). To elucidate the interactions of the water cycle with the biochemical, energy, and carbon cycles, numerous other variables are monitored (Valdes-Abellan et al., 2017). The key vegetation characteristics that are monitored include canopy height, leaf area index (LAI), leaf water potential, sap flow, rooting depth and distribution, plant water stress, canopy and/or vegetation water content, and temperature (Poyatos et al., 2021; Loritz et al., 2022; Zeng and Su, 2024). Eddy covariance measurements, some of which are connected through networks, such as FLUXNET, are used to obtain evapotranspiration and carbon fluxes at the local level. Sap flow sensors, some of which are organized in the SAPFLUXNET network (Poyatos et al., 2021), can be used to quantify transpiration rates. The use of tracer measurements, such as isotope and dye studies, enables the tracking

and differentiation of water fluxes (Klaus and McDonnell, 2013; Penna et al., 2018). Lysimeters are used to determine groundwater recharge and the associated concentrations of, e.g., nitrate at the point scale.

The use of uncrewed aerial systems (UASs; e.g., Dugdale et al., 2022; Romano et al. 2023) and satellite platforms (e.g., Durand et al., 2021; De Lannoy et al., 2022) for remote sensing has emerged as a valuable supplementary method in relation to ground-based observations in HOs for gathering information over large heterogeneous areas, as well as for upscaling or downscaling hydrological variables (e.g., McCabe et al., 2017; Manfreda et al., 2024; Su et al., 2020). Recently, higher-resolution observations of various hydrological variables have become available, including soil moisture (Han et al., 2023), snow depth (Lievens et al., 2022), and irrigation rate (Dari et al., 2023). These observations can be used together with coarser-scale products, including total water storage data from the Gravity Recovery and Climate Experiment (GRACE) mission and discharge data from the Surface Water and Ocean Topography (SWOT) mission. The deployment of multiple sensors, as seen in the various Sentinel and Landsat missions, can enhance the accuracy and resolution of the data. The European Space Agency (ESA) and the United States National Aeronautics and Space Administration (NASA) are engaged in collaborative efforts with public and private organizations to develop relevant new missions and to disseminate a range of products, including evapotranspiration estimates through the SEN-ET (Guzinski and Nieto, 2019; Guzinski et al., 2020) and OpenET (Melton et al., 2022) initiatives.

It is evident that the key to progress in hydrological understanding will be contingent upon the integration of these observation platforms. These platforms should integrate technologies such as remote sensing, high-performance-computing resources, artificial intelligence, and the Internet of Things while acknowledging the influence of geochemical and biotic heterogeneity, as well as of socioeconomic processes, on water and energy fluxes.

While observations are the cornerstone of progress in hydrological understanding (Sivapalan and Blöschl, 2017), models are equally essential for hypothesis testing and making predictions that are practically relevant (Brooks et al., 2015; Baatz et al., 2018; Bogena et al., 2018; Bechtold et al., 2019; Nearing et al., 2024). However, hydrological models, particularly those of a complex nature, frequently rely on lumped parameter calibration. This means that model parameters are adjusted based on aggregated (or lumped) fluxes, such as those observed in streamflow measurements at the outlet of the catchment. Although this approach can be effective, it can also result in limitations. A significant challenge is the assumption that the model's behavior is uniform across the entire catchment. This assumption might not hold true, especially in heterogeneous catchments with diverse topography, land uses, and soil types. In such cases, relying exclusively on lumped fluxes may result in suboptimal model

performance. An integrated observation approach enables the calibration based on insightful analyses of process complexity through systematic learning from distributed hydrometeorological data given that catchments are complex systems with structured heterogeneities, which give rise to non-linear interactions and feedbacks between the component processes (Vereecken et al., 2015; Li et al., 2022). One aspect of integration is the assimilation of observations into hydrological models (Mwangi et al., 2020; Kumar et al., 2022; De Lannoy et al., 2022) to estimate unobserved variables, improve predictions, and calibrate and validate satellite retrieval (Colliander et al., 2021). Paleo-reconstructions represent another example of integration and are instrumental in developing a more comprehensive understanding of how dynamic, abiotic, and biotic catchment characteristics co-evolved well before the advent of instrumental records (Troch et al., 2013). Climate shifts leave a multitude of signatures in the natural world, influencing processes such as tree growth and the distribution of plant species. The advent of increasingly sophisticated analytical techniques has facilitated a rapid growth in knowledge regarding past climate and river ecosystem variability. Of particular benefit are reconstructions of river flow and erosion derived from natural archives (Torbenson et al., 2021; Schöne et al., 2020; Strelnikova et al., 2023).

## 3   HO networks and hydrological synthesis

The sustainability of HOs is a matter of concern. Financial and logistical constraints have posed challenges to the long-term operation of HOs, jeopardizing essential maintenance, equipment upgrades, and personnel training. This ultimately compromises the quality and continuity of hydrological data collection and analysis. Data gaps and the lack of continuity in the data collection process hamper the identification and understanding of hydrological change, which represents a significant societal need for hydrology in the present and the future (Montanari et al., 2013). In light of the frequently constrained budgetary resources available for each site, many studies have focused on measuring lumped hydrological fluxes (e.g., the streamflow at the catchment outlet), while observatories that prioritize the analysis of spatial details remain relatively scarce (e.g., Blöschl et al., 2016). Site-specific methods, tailored to the site-specific UPH, have frequently resulted in advancements in the understanding of a specific hydrological process but have not fully exploited the potential for synergies with other HOs. Consequently, the outcome has frequently been an increase in fragmented knowledge rather than progress in understanding the interactions of hydrological processes, which is so urgently needed.

To address these issues, scientists have proposed initiatives to sustain long-term operation, harmonization, and standardization of both hydrometeorological data and ecohydrological models in HO networks (Zoback, 2001; Reid et al., 2010; Kulmala, 2018). In numerous instances, hy-

drological observations have been integrated into interdisciplinary research programs in terrestrial observatories, which are scientific facilities designed to observe and study various aspects of the Earth's surface, atmosphere, and interior. Terrestrial observatories collect data on a range of phenomena, including earthquakes, volcanic activity, weather patterns, climate change, and the movement of tectonic plates. Hydrological observations play a crucial role in the context of terrestrial observatories. Notable initiatives that have integrated existing environmental research infrastructures include the pan-European ENVRI initiative (https://envri.eu, last access: 2 February 2024) and the global GERI initiative (https://global-ecosystem-ri.org/, last access: 2 February 2024, Loescher et al., 2022). Networks such as FLUXNET (https://fluxnet.org, last access: 2 February 2024) and the Integrated Carbon Observation System (https://www.icos-cp.eu, last access: 2 February 2024) collect standardized data on the soil surface energy balance and evapotranspiration. The network of Critical Zone Observatories aims to understand critical-zone processes, with a particular focus on hydrologic monitoring (Brantley et al., 2017; Anderson et al., 2008; Gaillardet et al., 2018). The integrated European Long-Term Ecosystem, critical zone, and socio-ecological Research infrastructure (https://elter-ri.eu, last access: 2 February 2024) is establishing a network of approximately 200 integrated terrestrial observatories across Europe, with hydrological monitoring forming a component of this initiative. In the field of agriculture, the United States Department of Agriculture (USDA) is providing support for the Long-Term Agroecosystem Research (LTAR) initiative (https://ltar.ars.usda.gov/, last access: 2 February 2024), which combines strategic research projects with common measurements across multiple agroecosystems, including croplands, rangelands, and pasturelands.

The advent of digital technology and data exchange platforms has enabled scientists to aggregate and jointly analyze data streams from disparate locations in a manner that was previously unfeasible. This is contingent upon the standardization and harmonization of existing protocols and methods for hydrological observation. The extant research infrastructures have already established standards for the environmental variables they collect. The harmonization of such standards across disciplinary infrastructures represents a crucial building block toward enhanced integration and should be reflected in future strategies for designing international environmental research.

The cross-site synthesis of hydrological processes serves to fill the gap between site-specific studies and broader, more generalizable knowledge (Zacharias et al., 2024). The objective is to integrate information from multiple sites and sources to identify patterns, trends, and relationships that can lead to the development of a more robust and transferable body of knowledge for model development and, ultimately, more effective decision-making. The implementation

of cross-site synthesis typically entails the following steps, as illustrated in Fig. 1:

- formulating the UPH
- data collection by using standardized protocols
- use of community-shared hydrological models
- comparative hydrology
- meta-analyses to consolidate results.

The initial step is to formulate scientifically interesting questions that address existing knowledge gaps and contribute to a broader understanding and to societal benefits in the field of hydrology (see Appendix). To ensure consistency in data quality, it is essential to harmonize the measurement techniques and quality control protocols employed. Community networks and centralized data repositories can facilitate this process and provide access to standardized and curated datasets. Community-shared hydrological models can be employed to represent the complex interactions between hydrological processes, ecosystems, and human activities. The models are calibrated and tested using the harmonized data from multiple sites, thereby enhancing their predictive capabilities and generalizability. Notable initiatives are already operational, including the Unified Forecast System (UFS) which is a community-based, coupled, comprehensive Earth modeling system used for weather forecast applications (https://www.ufscommunity.org/articles/hierarchical-system-development-for-the-ufs/, last access: 2 February 2024). Comparative studies have been instrumental in identifying the key drivers of hydrological variability and in establishing generalizable principles. This is accomplished through a comprehensive and systematic comparison of hydrological processes and responses across a range of sites while accounting for several factors, including climate, topography, land use, and management practices. In addition, meta-analyses can synthesize and compare findings from multiple studies, identify recurrent patterns and trends in integrated measurements and model simulations, and present consolidated results in a coherent manner.

By following these steps, hydrologists can effectively implement cross-site synthesis, thereby advancing the field of hydrology toward a more generalizable and transferable body of knowledge. This can inform more effective decision-making with regard to the management of water resources and the adaptation to climate change in a variety of contexts.

Cross-site synthesis helps unveil hidden assumptions that may be embedded in site-specific studies, thereby enabling researchers to critically assess the validity of these assumptions and explore alternative perspectives. The identification of common principles and practices allows researchers to develop transferable knowledge that can be applied to other settings, thereby accelerating progress in research and practice. Examples of cross-site synthesis already exist in the literature. For example, Wlostowski et al. (2021) conducted a

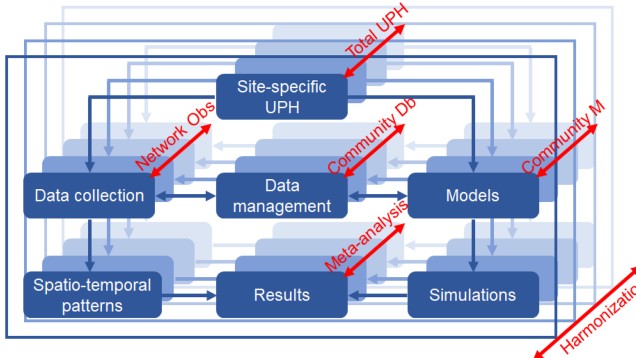

**Figure 1.** Schematic of the proposed cross-site synthesis. Obs, Db, M, and UPH indicate observatories, database, models, and Unsolved Problems in Hydrology, respectively.

meta-analysis of hydrologic signatures from 15 catchments in the US Critical Zone Observatory (CZO) network, which revealed consistent relationships between critical-zone structure and hydrologic response across sites. Similarly, Addor et al. (2018) examined the predictability of hydrologic signatures for the catchments included in the Camels dataset but found that the relationship between these signatures and catchment attributes other than climate characteristics was weak.

Comparative analyses have yielded a range of interesting results, although there is not yet complete concordance. For instance, some studies have indicated that afforestation may result in a decrease in water yield, whereas others have identified an increase. Two distinct theoretical frameworks have been put forth to explain the aforementioned conflicting results (Ellison et al., 2012). The *demand-side* perspective places emphasis on the increase in transpiration and the subsequent reduction in streamflow, particularly in catchments smaller than a few square kilometers (Schilling et al., 2008; Kim et al., 2013; Nasta et al., 2017). In contrast, the *supply-side* perspective posits that afforestation will intensify precipitation, thereby increasing streamflow, in downwind catchments (Ellison et al., 2012). Similarly, the impact of reforestation on dry-season flows is contingent upon the relative importance of increased infiltration and evapotranspiration rates (Bruijnzeel, 1989).

As demonstrated by the preceding case studies, cross-site synthesis provides a valuable approach for quantifying the spatial variability of hydrological processes and identifying consistent patterns in phenomena such as droughts and floods. These examples will ultimately inform water management practices around the world while maintaining the tracking and awareness of local hydrological particularities.

## 4 How to manage a network of hydrological observatories

For the sake of the argument, we assume that a fixed community budget has been allocated for the establishment and operation of a hypothetical network of HOs in the European Union (EU). Two potential scenarios can be envisaged. In the first scenario (scenario 1), the available financial resources are distributed among a multitude of moderately instrumented HOs situated throughout the EU, with the objective of addressing challenges in hydrology with extensive geographical coverage. Figure 2 illustrates an example of a moderately instrumented site belonging to a hydrological observatory network in scenario 1. This plan reflects the current status of the majority of HO networks around the world. The principal benefit of this approach is that the HOs are widely and effectively distributed, enabling the identification of cause-and-effect relationships and supporting the cross-site synthesis of the lumped hydrological responses (e.g., rainfall–runoff relationship, Budyko analysis) across diverse continental landscapes (Wagener et al., 2007; Ehret et al., 2014; Jones et al., 2012; Kuentz et al., 2017; Templer et al., 2022).

In scenario 1, a combination of centralized and distributed components is utilized. Distributed components provide observed data that are managed by different entities (e.g., universities, research institutions, government agencies) across geographically spread sites. To guarantee the comparability of data, it is essential to implement standardized protocols for data collection, storage, quality assurance, and analysis. This will alleviate the burden associated with the cross-site synthesis. Centralized data management facilitates the accessibility of data across multiple sites. Furthermore, additional central thematic elements may be provided, such as those pertaining to communication and knowledge transfer or those relevant to modeling applications. The organizational structure may be based on other successfully established or planned distributed continental infrastructures. Notable examples include ICOS (Integrated Carbon Observation System) and eLTER (integrated European Long-Term Ecosystem, critical zone, and socio-ecological Research infrastructure). Free data availability and accessibility of the sites should be a fundamental aspect of the scenario designs.

In scenario 1, collaboration and partnership among different stakeholders are crucial. Such collaboration may facilitate broader opportunities for citizen and stakeholder participation, particularly given the distributed nature of the scenario and the encouragement of local initiatives.

In the second scenario (scenario 2), research efforts and financial resources are pooled into a limited number of pilot HOs, each equipped with massive instrumentation. Similar initiatives can be found in related fields of study. In oceanography, a limited number of costly research vessels are made available, primarily through the financial support of national governments. This approach enables numerous

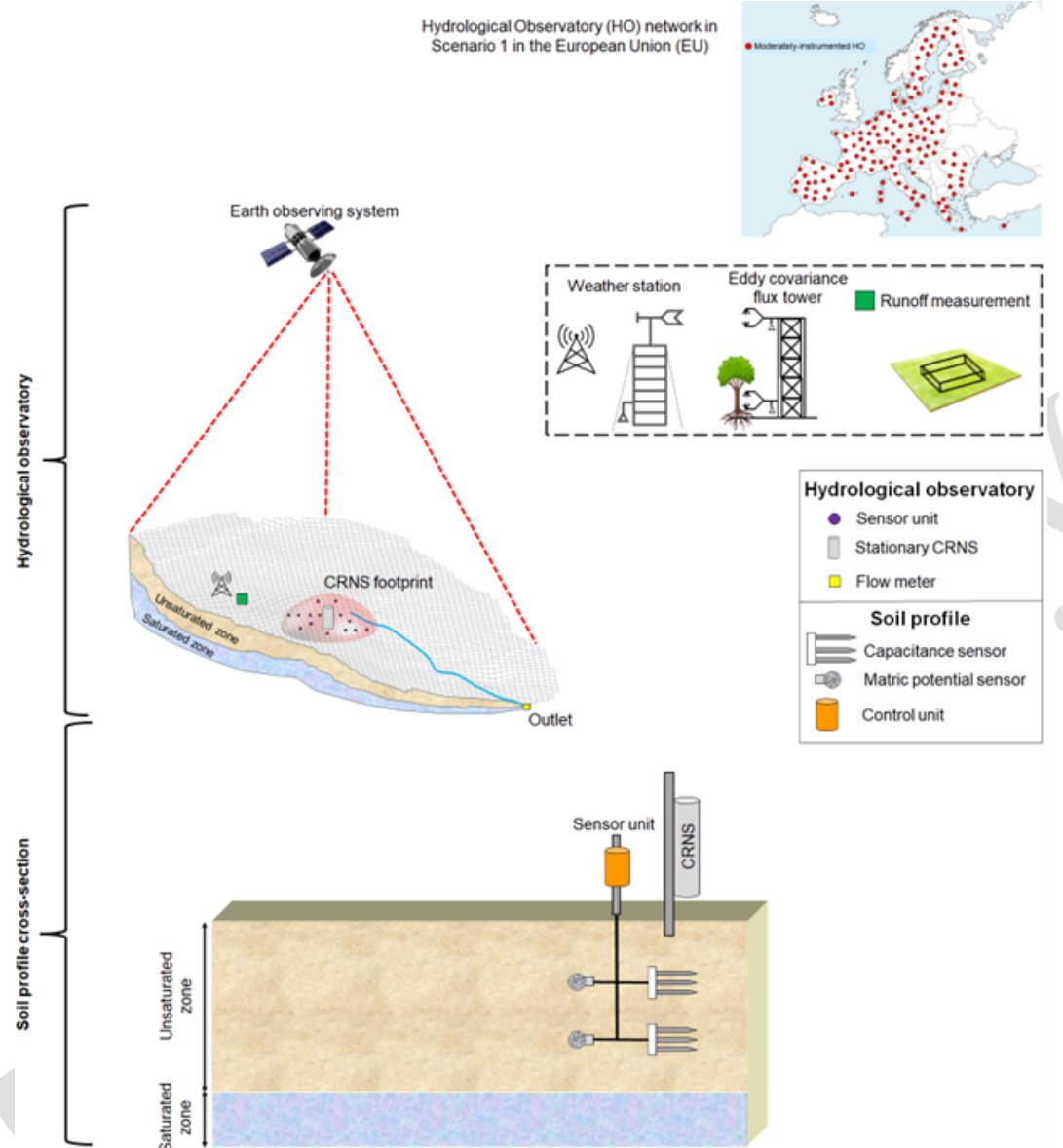

**Figure 2.** Graphical illustration of a hydrological observatory (HO) network in the European Union (EU) in scenario 1. Each sub-catchment is equipped with basic instrumentation: a weather station, a runoff gauging station, a cosmic-ray neutron sensor (CRNS) with a wireless sensor network controlling soil profile sensors, and a streamflow sensor at the catchment's outlet. Satellite products are available anywhere in the world. The soil profile cross-section illustrates the soil profile sensor unit and the stationary CRNS.

researchers to engage in collaborative community experiments, thereby facilitating a deeper understanding of specific oceanic regions. One illustrative example is the Multidisciplinary drifting Observatory for the Study of the Arctic Climate (MOSAiC), which undertook a drift over Arctic sea ice from October 2019 to September 2020 aboard the Polarstern research vessel (Rabe et al., 2022). In the field of geology, the cost of drilling into the Earth is almost equally expensive. The consolidation of resources permits geologists from diverse geographical locations to engage in collaborative drilling programs, such as the International Continental Scientific Drilling Program (ICDP). Of course, in both cases, the research questions or aims are explicitly delineated. In the case of MOSAiC, the aim was to gain a deeper understanding of the influence of the Arctic on the global climate, given that the Arctic has experienced a more pronounced warming trend than any other region of the world. Concerning ICDP, the objective was to gain a deeper understanding of the Earth's processes and structure at the most interesting locations. In both instances, participation is contingent upon the successful completion of an application and review process overseen by an international committee.

Similarly, a small number of HOs equipped with comprehensive instrumentation and managed by an international team of experts from various disciplines could represent the pinnacle of hydrological field research. Figure 3 shows a hypothetical supersite established along an ideal transect within the European Union (scenario 2). A high-density network of sampling and monitoring units for soil hydrology research is designed and planned for each supersite. This infrastructure, as yet unrealized, would facilitate a comprehensive understanding of water dynamics in the groundwater–soil–plant–atmosphere continuum and of water circulation in the surface and subsurface domains within a few sites on each continent. In this case, cross-site synthesis would support the application and refinement of complex hydrological models based on fundamental insights into complex, non-linear processes that are modulated by scale-dependent feedbacks and multiscale spatiotemporal heterogeneity.

A wealth of data would enable an unprecedented unraveling of hydrological processes at the hillslope and/or catchment scale based on observations of water and energy fluxes in the groundwater–soil–vegetation–atmosphere continuum at high spatial and temporal resolutions.

In scenario 2, the research questions should be presented boldly (see discussion in Davis, 1926; Beven and Germann, 2013; McDonnell, 2014; Burt and McDonnell, 2015; Kirchner, 2016; Blöschl et al., 2019; Gao et al., 2023). Some interesting examples of scenario 2 have already been documented in controlled environments. Biosphere-2 (B2) in Tucson, Arizona (Evaristo et al., 2019), is a research facility comprising a tropical rainforest biome, a mesocosm enclosed in a pyramidal glass structure. Additionally, the Landscape Evolution Observatory (LEO) comprises three artificial hillslopes equipped with a dense network of soil sensors. The observatory is focused on understanding the interaction between water and weathering processes (Van Den Heuvel, 2018; Bauser et al., 2022). Another example is the artificial Chicken Creek catchment in Germany, which has served as the fulcrum of comparative community research on runoff generation (Holländer et al., 2009).

Once more, as with the sister disciplines, the choice of location is of the utmost importance. The selected locations should represent hydrological situations that are particularly conducive to addressing the primary research question. The Austrian Hydrological Open Air Laboratory (HOAL) (Blöschl et al., 2016) was designed with the specific objective of facilitating a more comprehensive understanding of rainfall–runoff processes. It is ideally suited for this purpose, featuring a range of different runoff generation processes, including surface runoff, springs, tile drains, and wetlands. Another example is provided by the Alento Hydrological Observatory, which aims to elucidate the effects of the typical Mediterranean seasonality of climate, as well as the effects of land use and/or land cover changes on water flow in the critical zone of a representative southern European catchment (Nasta et al., 2017; Romano et al., 2018). To explore land–atmosphere feedbacks, it is recommended that a catchment of considerable size be selected (Späth et al., 2023). In addition, when selecting a location for a new, high-budget HO, it may be beneficial to consider the existence of so-called environmental archives in the area of interest. The ease of accessibility and the availability of infrastructure may be other factors to consider, but this could result in a geographic and climatic bias of the research sites.

Few supersites would require a central governing body that would likely be responsible for overseeing all aspects of the supersites, including instrument deployment and maintenance, as well as data collection and analysis. Such an entity could be a dedicated government agency with a specific mandate or a research consortium with significant resources. The establishment of a single entity in charge, operating as a central authority, would facilitate the decision-making process with regard to instrument upgrades, research focus, and site and data access.

Super-sites with advanced instrumentation might attract highly specialized researchers, leading to a concentration of expertise in specific areas. The implementation of standardized sensors would result in cost savings and enhanced efficiency in the collection and processing of data. Conversely, the specific hydrological environment may require the use of specialized instrumentation or measurement techniques. A lack of flexibility in standardization can limit the ability to adapt to new research questions or emerging challenges. Notable examples of standardization efforts include the Global Network of River Observatories (GLORIA) and the World Meteorological Organization (WMO) guidelines for hydrological stations. By taking these factors into careful consideration and adopting a balanced approach, hydrological observatories can harness the power of standardization while maintaining flexibility and adaptability. To ensure equity and to stimulate greater involvement in scenario 2, it is essential to establish a collaborative governance structure that incorporates a diverse range of stakeholders in decision-making processes pertaining to supersite operations and data utilization. The governance and site access aspects are well presented in initiatives such as the International Continental Scientific Drilling Program (ICDP), which addresses geodynamic processes, solid Earth geohazards, sustainable geo-resources, and environmental change (https://www.icdp-online.org/about-icdp/entities/, last access: 2 February 2024). Another noteworthy example is the Alfred Wegener Institute (AWI), which aims to understand the complex processes in the Earth system and the impact of global warming on the oceans and polar regions (https://www.awi.de/en/, last access: 2 February 2024). The AWI maintains a network of well-instrumented long-term observatories, comprising both stationary devices and mobile components that are used for studies pertaining to oceanography, meteorology, and geophysics (https://www.awi.de/en/expedition/observatories.html, last access: 2 February 2024).

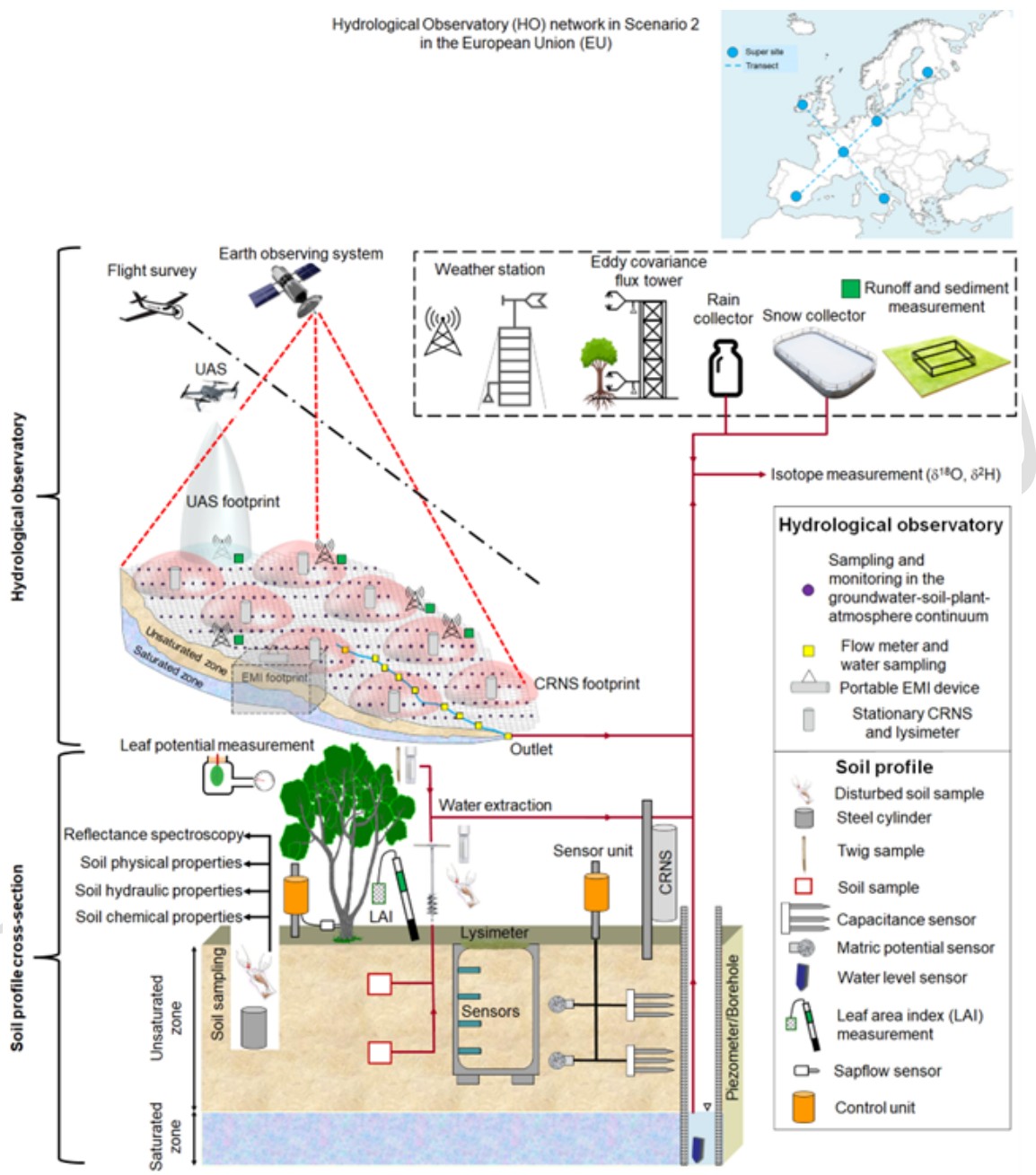

**Figure 3.** Graphical illustration of a hydrological observatory (HO) network in the European Union (EU) in scenario 2. Each sub-catchment established along an ideal transect is equipped with a high-density network of sampling and monitoring units for soil hydrology research. Frequent uncrewed aerial system (UAS) and aircraft surveys are organized over the experimental area. Satellite products are available anywhere in the world. Frequent campaigns of geophysical (electromagnetic induction, EMI technique) and tracing (stable isotopes in water, such as $\delta^2 H$ and $\delta^{18} O$) measurements are carried out across the HO. Flow monitoring and water sampling are carried out along the stream. The soil profile cross-section shows the monitoring and sampling activities in the groundwater–soil–plant–atmosphere continuum in a position of the dense point grid (purple circles).

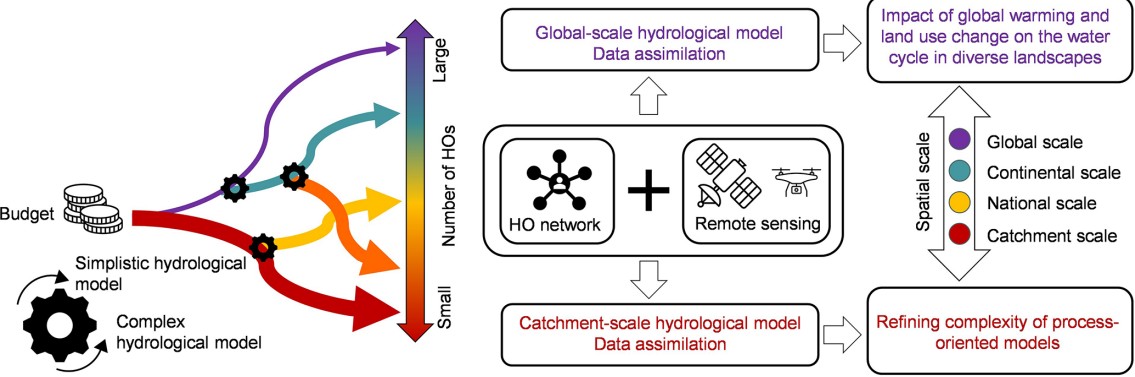

**Figure 4.** Possible configurations of hydrological observatory (HO) networks are illustrated, spanning a range from a few (color-coded in red) to numerous (color-coded in blue) HOs. The thickness of the arrows indicates the quantity of instruments present in each HO. The data obtained from the HO network and remote sensing platforms are used to inform hydrological models of different complexities, enabling us to address specific scientific questions across disparate spatial scales.

The selection of the optimal scenario is contingent upon the research questions deemed most pertinent, the capacity to secure funding, and the extent to which the hydrological community is willing and able to collaborate. In the context of financial constraints, no single alternative can be considered to be inherently superior. A distributed network of numerous HOs is well-suited to broad-scale inquiries, whereas a network of a few supersites is particularly adept at facilitating in-depth process understanding.

A potential way forward with regard to the aforementioned issues could be to merge the two scenarios into a dynamic or adaptive hybrid approach (see Fig. 4). The establishment of a network of geographically distributed observatories would facilitate the achievement of a high level of representativeness regarding existing gradients of geology, climate, and land use. This would assist in the identification of priority areas requiring further investigation and would alleviate some of the bias in current hydrological studies (Burt and McDonnell, 2015; Tarasova et al., 2024). The development of some of these observatories into hydrological supersites, contingent on the availability of resources, would allow for the investigation of specific topics, such as karst hydrology, water scarcity, floodplains, forest hydrology, precision agriculture, and different runoff generation mechanisms. This approach enables targeted investigations at specific locations with high-resolution data, which can then be exploited to support the development of high-fidelity models.

It is similarly feasible to reverse this scenario. Should a supersite situated in a particular bioclimatic zone yield scientific breakthroughs, it may be possible to establish a network of HOs in regions exhibiting analogous hydrological behavior. The key factor is to leverage the strengths of each approach while operating within the confines of the allocated budgetary resources. By integrating the strengths of the two approaches, one can attain a balance between representativeness (a distributed network) and a detailed understanding (supersites). This approach ensures the optimal exploitation of resources while maximizing the scientific output.

## 5 Concluding remarks

To address water-related issues at the catchment scale across diverse global contexts, it is imperative to develop adaptation and mitigation strategies that are grounded in evidence gathered by HOs. The previous situation, which was characterized by a myriad of relatively unconnected, moderately and differently instrumented HOs that were supported by grant-to-grant funding, has resulted in significant but fragmented knowledge. This has impeded comparative studies and has hindered scientific progress. New initiatives are being proposed with the objective of enhancing the coordination of HO networks, thereby enabling efficient cross-site synthesis. In light of financial constraints, we need to identify a common vision for the optimal allocation of resources.

A network of numerous HOs provides broad spatial coverage, enabling the capture of variations in environmental conditions across diverse regions, ecosystems, and land uses. Environmental change can manifest itself differently across regions due to the influence of local climate, geography, and human activities. A network of numerous observatories offers the opportunity to monitor these interactions and to capture feedbacks, teleconnections, and cross-scale dynamics that may not be observable at individual observatories. In contrast, a limited number of intensively instrumented observatories permit the collection of high-resolution data, the testing of novel hypotheses, and the informing of comprehensive process-oriented hydrological models. This choice can capture variations at smaller spatial and finer temporal scales, thereby providing a more nuanced understanding of environmental and hydrological processes. Nevertheless, the strategy of pooling all financial efforts into a limited number of intensely instrumented hydrological observatories will exac-

erbate the issue of knowledge transferability and geographic bias in hydrological data and understanding. It is therefore necessary to devise strategies for generalization, potentially drawing inspiration from the Prediction in Ungauged Basins initiative.

In the context of accelerated global transformation, there is a pressing need to establish a network of HOs. The question of how to organize and manage such a global network, including the number of observatories, remains a topic of discussion. Both distributed networks and super-sites offer valuable contributions to the advancement of hydrological understanding. We envision a dynamic hybrid approach that combines the two aforementioned visions in a manner that does not exclude either of them from consideration. It is important to raise public awareness about the significance of hydrological research and its linkages with a multitude of other disciplines, including atmospheric science, soil science, biochemistry, pedology, ecology, microbiology, geology, plant physiology, and remote sensing. Such an approach can garner support and increase funding opportunities. It is our hope that all hydrologists will engage in a discussion process with the aim of refining and building upon the ideas presented in this paper.

**Appendix A**

**Table A1.** The 8th Galileo Conference "A European Vision for Hydrological Observations and Experimentation" was held in Naples (Italy) on 12–15 June 2023. Following presentations and discussions, we report the most intriguing questions in hydrology that emerged from the conference. Additionally, we conducted a literature review and identified several key points that warrant further investigation.

| | Research questions in scenario 1 |
|---|---|
| 1 | How can we improve the quantity and quality of measurements in data-poor regions? |
| 2 | Where and how can we deploy the sensors to get the most information without wasting too much effort? |
| 3 | Are measurements taken in the past still valid in the future? How will accuracy or precision change with technological advances? Do we need to remove all "inaccurate" historical data and keep only "currently accurate" data? Is the assumption of a steady hydrological system valid? Can we simplify the system by linearizing a non-linear system behavior? |
| 4 | What roles do continuous and ephemeral waterbodies, including ponds, lakes, rivers, streams, marshes, swamps, etc., play in influencing water quantity and quality in the catchment? |
| | Research questions in scenario 2 |
| 1 | What are the hydrologic laws at the catchment scale? How do they change with scale? |
| 2 | How can we use innovative technologies to measure surface and subsurface properties, states, and fluxes at a range of spatial and temporal scales? |
| 3 | How can different multi-scale observations be assimilated into a hydrological model to improve model predictability? |
| 4 | How do we obtain large-scale flux measurements and feedbacks to analyze the water dynamics within and between the compartments of the groundwater–soil–plant–atmosphere continuum? |
| 5 | How is the water cycle influenced by the other cycles (carbon, nitrogen, etc.)? |
| 6 | How can the dynamics and feedbacks at groundwater–soil, groundwater–surface water, soil–plant, soil–atmosphere, and plant–atmosphere interfaces be assessed? |
| 7 | How do we incorporate plant physiological aspects into hydrological models? |
| | Research questions in both scenarios |
| 1 | What is the impact of preferential flow on catchment-scale water flow dynamics? |
| 2 | How can remote sensing provide more reliable information on soil moisture, changes in water storage, surface energy balance, and evapotranspiration at appropriate spatial and temporal scales (Lettenmaier et al., 2015)? |
| 3 | What causes spatial heterogeneity and homogeneity in runoff, evapotranspiration, subsurface water, and material fluxes (carbon and other nutrients, sediments) and in their sensitivity to their controls (e.g., snowfall regime, aridity, response coefficients)? |
| 4 | How can hydrological models be adapted to be able to extrapolate changing conditions, including changing vegetation dynamics? |
| 5 | How can we disentangle and reduce model structure, parameter, or input uncertainties in hydrological prediction? |
| 6 | Is it better to emphasize uncertainty or causality? |
| 7 | How do vegetation types, distributions, and dynamics shape hydrological processes, particularly in terms of water quality, water quantity, and energy fluxes at the catchment scale? |
| 8 | How can we integrate the different spatial and temporal scales of observations, processes, and models? |
| 9 | How can we develop socio-hydrological models by allowing for anthropogenic disturbances in the ecosystem? |

*Code and data availability.* No code or datasets were used in this paper. CE1

*Author contributions.* PN and NR were responsible for the initial drafting of the paper, as well as its subsequent review and editing. All authors were involved in the conceptualization of the opinions expressed in this paper and contributed to the writing of the final version of the paper.

*Competing interests.* At least one of the (co-)authors is a member of the editorial board of *Hydrology and Earth System Sciences*. The peer-review process was guided by an independent editor, and the authors also have no other competing interests to declare.

ther geographical representation in this paper. While Copernicus Publications makes every effort to include appropriate place names, the final responsibility lies with the authors.

*Acknowledgements.* This paper is a result of the presentations and discussions held at the 8th Galileo Conference, entitled "A European Vision for Hydrological Observations and Experimentation", which took place in Naples, Italy, from 12 to 15 June 2023. In his keynote lecture, entitled "The Future of Hydrology: Nature or Nurture", presented on 14 June 2023, Günter Blöschl discussed the concept of focusing research efforts on a limited number of catchments worldwide. This prompted a lively exchange of opinions among the attendees. The USDA is an equal-opportunity provider and employer. Salvatore Manfreda acknowledges the RETURN Extended Partnership which received funding from the Next-Generation EU (National Recovery and Resilience Plan – NRRP, Mission 4, Component 2, Investment 1.3 =- D.D. 1243 2/8/2022, grant no. PE0000005). Paolo Nasta and Nunzio Romano acknowledge the partial support provided by the Ministry of University and Research (MiUR) through the MiUR-PRIN-PNRR-2022 project entitled "ASAP: Assessing and mapping novel agroecosystem vulnerability and resilience indicators in southern Italy" (grant no. P20228L3KX). The authors would also like to express their gratitude to the executive editor in charge, Thom Bogaard, and the two reviewers (Andrew Guswa and an anonymous reviewer) for their constructive comments, which have contributed to enhancing the quality of this paper.

*Financial support.* This study reported in this paper was supported by the MiUR-PRIN-PNRR-2022 project "ASAP: Assessing and mapping novel agroecosystem vulnerability and resilience indicators in southern Italy" (grant no. P20228L3KX).

*Review statement.* This paper was edited by Thom Bogaard and reviewed by Andrew Guswa and one anonymous referee.

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

**Remarks from the language copy-editor**

CE1    Please note the slight edits.

**Remarks from the typesetter**

TS1    We will ask the handling editor for approval in this case. Please prepare an explanatory document (pdf) with the new figure which we can send to the editor via our system to ask for approval.