# Peer review of "HESS Opinion paper: 1 2"

_EGUsphere, 2024_

## Author Comment (AC1)

**HESS Opinion paper:**
**Towards a common vision for the future of hydrological observatories**

**EDITOR**

We would like to thank the Editor for handling our contribution. The first reviewer provided very positive and extremely constructive comments, which helped us substantially improve the quality of our manuscript. We have modified some parts of the manuscript in accordance with his suggestions. The second reviewer raised concerns, especially regarding the initial section of the manuscript. Therefore, we have removed the first two figures (which actually were not well received by both reviewers) from the second section and expanded the last part of the manuscript pertaining to the comparison of two hypothetical scenarios for managing hydrological observatories (HOs). In response to the first reviewer's suggestion, we have added two new figures (Figures R1 and R2 in this reply letter) to Section #4 to provide further emphasis on the comparison of the two management scenarios.

We have revised the original manuscript submitted to Hydrology and Earth System Sciences, and we do hope that the new version addressed the majority of the reviewers' concerns. For reference, we have included line numbers relevant to the manuscript without tracked changes.

**Reviewer 1**

Review by Andrew J. Guswa, Smith College, aguswa@smith.edu

This editorial, which grew out of discussions at the 8th Galileo Conference held in Napoli, Italy in June 2023, raises the question of how funding and resources for hydrological observatories could best be deployed, considering two scenarios: 1) a large number of moderately instrumented sites or 2) a small number of highly instrumented "super sites" (the dynamic-response approach notwithstanding). The question is a provocative and important one. With the current version of the manuscript, I also found myself wishing for more details of the two models, more discussion of the scientific tradeoffs between them, and perhaps considering ideas beyond the two. Additionally, I think it would be interesting and valuable to discuss potential differences in governance structures, data access, and equity issues. To make room for such detail and discussion, I think some of the preamble/motivation could be shortened and one or two of the figures removed/replaced.

**REPLY**: We would like to thank Ref. #1 (A. Guswa) for reviewing our work. In the following sections, we addressed his concerns and integrated his suggestions where appropriate. In this response, line numbers refer to the manuscript without using tracked changes. We have removed any elements that might have caused confusion and clarified several points.

Specifically, perhaps Figure 1 could be expanded into two – one that presents the instruments and their distribution associated with a site for Scenario 1, and a second figure that presents the vision for a super-site associated with Scenario 2. Would they have the same types of instruments, but with different levels of intensity and resolution? Or would there be additional variables and characteristics measured in the super-sites? Providing more concrete detail – even if purely hypothetical – will help the reader better understand the two models and the differences between them.

**REPLY:** Before illustrating the main differences between the two management scenarios, a hydrological observatory is defined as the cyber-physical infrastructure established within a catchment to monitor the hydrological variables and fluxes and to characterize the hydrological behavior of the 3D spatial domain. The catchment is assumed to be the fundamental hydrological unit with well-defined system boundaries. It is from this unit that the impact of anthropogenic disturbances (global warming, land use change, aquifer contamination, etc.) on water resources can be evaluated through a long-term data analysis. Given the impracticality of full catchment coverage, the hydrological observatory concentrates on a cluster of representative sub-catchments (spatial resolution of hectares) which are representative of land use,

geomorphology, topography, and pedology similarities (Bogena et al., 2006). Therefore, the selected sub-catchments are equipped with wireless sensor networks for continuous data collection and subjected to different field campaigns depending on budget constraints.

As described in the original manuscript (Section 4), we have assumed that a fixed community budget has been allocated for the establishment and operation of a hypothetical network of HOs in the European Union (EU). Figure R1 shows an example of a moderately-instrumented site belonging to a hydrological observatory network in Scenario 1. This plan reflects the current situation at most HO networks around the world. S1 offers the clear advantage of widely distributed HOs across each continent, enabling cross-site synthesis of the lumped hydrological response (e.g., rainfall-runoff relationship, Budyko analysis) across diverse continental landscapes.

[Figure]

*Figure R1. Graphical illustration of a hydrological observatory (HO) network in the European Union (EU) in Scenario 1. Each sub-catchment is equipped with basic instrumentation: a weather station, a runoff gauging station, a Cosmic-Ray Neutron Sensor (CRNS) with a wireless sensor network controlling soil profile sensors, a streamflow sensor at the catchment's outlet. Satellite products are available anywhere in the world. The soil profile cross-section illustrates the soil profile sensor unit and the stationary CRNS.*

Figure R2 delineates a hypothetical super-site established along an ideal transect in the European Union (Scenario 2). Each super-site should have a high-density network of sampling and monitoring units for soil hydrology research. This infrastructure, as yet unrealized, would facilitate a comprehensive understanding of water dynamics in the groundwater-soil-plant-atmosphere continuum and water circulation in the surface and subsurface domains within a few sites in each continent. In this case, cross-site synthesis would support the application and refinement of complex hydrological models based on fundamental insights into complex, non-linear processes modulated by scale-dependent feedback and multiscale spatio-temporal heterogeneity.

[Figure]

*Figure R2. Graphical illustration of a hydrological observatory (HO) network in the European Union (EU) in Scenario 2. Each sub-catchment established along an ideal transect is equipped with high-density network of sampling and monitoring units for soil hydrology research. Frequent unmanned aerial system (UAS) and aircraft surveys are organized over the experimental area. Satellite products are available anywhere in the world. Frequent campaigns of geophysical (electromagnetic induction, EMI technique) and tracing (stable isotopes in water such as $\delta^2H$ and $\delta^{18}O$) measurements are carried out across the HO. Flow monitoring and water sampling are carried out along the stream. The soil profile cross-section shows the monitoring and sampling activities in the groundwater-soil-plant-atmosphere continuum in a position of the dense point grid (purple circles).*

We thoroughly revised Section 4 by following this suggestion.

Then, given those differences, which of the UPH would be more amenable to Scenario 1 versus Scenario 2? The Appendix provides a list of important and intriguing questions – which would be better addressed by which Scenario? The manuscript already calls out differences in representativeness: e.g., Scenario 1 can cover more of "existing gradients of geology, climate, and land-use," whereas Scenario 2 would achieve "high spatial and temporal resolutions." More directly connecting these differences to the important questions would be a valuable enhancement to the paper.

**REPLY**: In our view, all of the UPH mentioned below (and many others) are susceptible to both scenarios, which are not mutually exclusive but rather mutually reinforcing concepts. For example, detailed investigations necessitating unparalleled instrumentation in S2, would facilitate the transfer of acquired knowledge to other biogeographical regions in S1. We aim to clarify this point in the revised article.

To reply to this reviewer, we grouped the UPH according to the associated Scenario:

**Questions in Scenario 1**

| | |
|---|---|
| 1 | How can we improve the quantity and quality of measurements in data-poor regions? |
| 2 | Where and how can the sensors be allocated to get full information without wasting excessive effort? |
| 3 | Are measurements taken in the past still valid in the future? How about accuracy/precision change with technological advancements? Do we need to remove all "inaccurate" historical data and keep only "currently accurate" data? Is the assumption of a steady hydrological system valid? Can we simplify the system by linearizing a nonlinear system behavior? |
| 4 | What role(s) do continuous and ephemeral water bodies, including ponds, lakes, rivers, streams, marshes, swamps, etc. influencing watershed water quantity and quality? |

**Questions in Scenario 2**

| | |
|---|---|
| 1 | What are the hydrologic laws at the catchment scale, and how do they change with scale? |
| 2 | How can we use innovative technologies to measure surface and subsurface properties, states and fluxes at a range of spatial and temporal scales? |
| 3 | How can various multi-scale observations be assimilated into a hydrologic model to enhance model predictability? |
| 4 | How do we get large-scale flux measurements and feedback to analyze the water dynamics within and between the compartments of the groundwater-soil-plant-atmosphere-continuum? |
| 5 | How is the water cycle influenced by the other cycles (carbon, nitrogen, etc.)? |
| 6 | How can the dynamics and feedback at groundwater-soil, groundwater-surface water, soil-plant, soil-atmosphere, and plant-atmosphere interfaces be assessed? |
| 7 | How do we include plant physiological aspects in hydrological models? |

**Questions in both scenarios**

| | |
|---|---|
| 1 | What is the impact of preferential flow on catchment-scale water flow dynamics? |
| 2 | How can remote sensing provide more reliable information on soil moisture, changes in water storage, surface energy balance, and evapotranspiration at suitable spatial and temporal scales (Lettenmaier et al., 2015)? |
| 3 | What causes spatial heterogeneity and homogeneity in runoff, evapotranspiration, subsurface water and material fluxes (carbon and other nutrients, sediments), and in their sensitivity to their controls (e.g., snowfall regime, aridity, reaction coefficients)? |
| 4 | How can hydrological models be adapted to be able to extrapolate changing conditions, including changing vegetation dynamics? |
| 5 | How can we disentangle and reduce model structural/parameter/input uncertainty in hydrological prediction? |
| 6 | Is it better to give more importance to uncertainty or causality? |
| 7 | What is the role of vegetation in the catchment? |
| 8 | How can we integrate the different spatial and temporal scales of observations, processes, and models? |
| 9 | How can we develop socio-hydrological models by considering anthropogenic disturbances in the ecosystem? |

We highlight the predominant number of UPH valid for both scenarios.

Additionally, I found myself wondering about the associated tradeoffs in governance, equity, site access, and data availability between the two scenarios. Perhaps this is not the place for such discussions; nevertheless, if possible, I think a modest discussion of those issues would be interesting and useful. What processes could/would be put in place to encourage participation from a broad range of stakeholders from the hydrologic community, particularly for Scenario 2?

**REPLY**: Thank you for this valuable comment. In the revised version, Section 4 has been expanded to include a more detailed discussion of the trade-offs between the two scenarios in terms of governance, site access, and equity.

**Scenario 1 (S1):** S1 relies on a combination of centralized and distributed components. Distributed components provide observed data managed by different entities (e.g., universities, research institutions, government agencies, etc.) across geographically spread sites. To ensure data comparability, it is essential to implement standardized protocols for data collection, storage, quality assurance and analysis. This will mitigate the burden of the cross-site synthesis. Centralized data management facilitates the accessibility of data across multiple sites. In addition, additional central thematic elements can be provided, such as those pertaining to communication and knowledge transfer, or those relevant to modeling applications. The organizational structure can be based on other successfully established or planned distributed continental infrastructures. Notable examples include ICOS (Integrated Carbon Observation System) or eLTER (Integrated European Long-Term Ecosystem, Critical Zone and socio-ecological Research Infrastructure). Free data availability and accessibility of the sites should be a fundamental aspect of the scenario designs. Collaboration and partnership among different stakeholders are crucial in S1 that might provide broader opportunities for citizen and stakeholder participation, particularly given the distributed nature of the scenario and the encouragement of local initiatives.

**Scenario 2 (S2):** A few super-sites would require a central governing body that would likely be responsible for overseeing all aspects of the super-sites, including instrument deployment and maintenance, as well as data collection and analysis. Such an entity could be a dedicated government agency with a specific mandate or a research consortium with substantial resources.
The establishment of a single entity, operating as a central authority, would facilitate a more streamlined decision-making process regarding instrument upgrades, research focus, and site and data access.
Super-sites equipped with advanced instrumentation might attract highly specialized researchers, leading to a concentration of knowledge and experience in specific hydrological domains. The implementation of standardized sensors would result in cost savings and improved efficiency in data collection and processing. In contrast, different hydrological environments may require specialized instrumentation or measurement techniques. A lack of flexibility in standardization may impede the ability to adapt to new research questions or emerging challenges. Examples of standardization efforts include the Global Network of River Observatories (GLORIA; https://www.gloria.ac.at) and the World Meteorological Organization (WMO; https://community.wmo.int/en) guidelines for hydrological stations. By carefully considering these factors and adopting a balanced approach, hydrological observatories can harness the benefits of standardization while maintaining flexibility and adaptability. To ensure equity and encourage greater participation in S2, it is essential to establish a collaborative governance structure that incorporates a diverse range of stakeholders in decision-making processes related to super site operations and data utilization. The governance and site access aspects are well presented in initiatives such as the International Continental Scientific Drilling Program (ICDP), which addresses geodynamic processes, solid Earth geohazards, sustainable georesources, and environmental change (https://www.icdp-online.org/about-icdp/entities/). Another relevant example is the Alfred Wegener Institute (AWI), which aims to understand the complex processes in the Earth system and the impact of global warming on the oceans and polar regions (https://www.awi.de/en/). The AWI maintains a network of well-instrumented long-term observatories, comprising both stationary devices and mobile components that are employed for studies related to oceanography, meteorology, and geophysics (https://www.awi.de/en/expedition/observatories.html).

Relatedly, while the authors position their two scenarios as end-members, I could imagine a scenario further in the direction of Scenario 1 (perhaps Scenario 0), in which instrumentation is deployed in a purely opportunistic way, taking advantage of construction projects associated with infrastructure upgrades. For example, one could imagine a policy that stipulates that any time a culvert is rebuilt or rehabilitated (e.g., in response to increasing storm intensity), a suite of monitoring instruments must also be installed. This would significantly reduce the upfront costs associated with hydrologic observation. The locations of such added monitoring would be far from planful, but there might be advantages in the sheer number of sites,

and ongoing advances in data handling/storage and machine-learning and data-science tools could facilitate new insights.

**REPLY**: We agree that by leveraging infrastructure upgrades for hydrological monitoring offers great opportunities to significantly reduce initial costs compared to the construction of new observation sites from scratch. Nevertheless, the opportunistic approach is applicable to all scenarios.

To make space for those expanded discussions, I think either Figure 2 or Figure 3 could be removed. Those schematics are nice, but perhaps are not necessary for communicating the central ideas of the paper (vs. the Graphical Abstract, which really presents the Scenarios in a compelling way).

**REPLY**: We concur with this comment and decided to remove the first two figures while expanding the discussion in the last part. Here, we added the new figures presented in the reply letter (R1 and R2 in this reply letter). The new figures illustrate a hypothetical super-site in comparison to a moderate site.

In summary, I appreciate the editorial as a provocation for the hydrologic community to consider and discuss the vision for hydrologic observations. I recommend expanding the comparison of the two scenarios to help the reader better understand the tradeoffs between them.

**REPLY**: We concur with this assessment and will therefore shorten the initial section as much as possible while expanding the discussion on the trade-offs, including the issues of governance, site access, and equity.

---

## Author Comment (AC2)

**HESS Opinion paper:**
**Towards a common vision for the future of hydrological observatories**

**EDITOR**

We would like to thank the Editor for handling our contribution. The first reviewer provided very positive and extremely constructive comments, which helped us substantially improve the quality of our manuscript. We have modified some parts of the manuscript in accordance with his suggestions. The second reviewer raised concerns, especially regarding the initial section of the manuscript. Therefore, we have removed the first two figures (which actually were not well received by both reviewers) from the second section and expanded the last part of the manuscript pertaining to the comparison of two hypothetical scenarios for managing hydrological observatories (HOs). In response to the first reviewer's suggestion, we have added two new figures (Figures R1 and R2 in this reply letter) to Section #4 to provide further emphasis on the comparison of the two management scenarios.

We have revised the original manuscript submitted to Hydrology and Earth System Sciences, and we do hope that the new version addressed the majority of the reviewers' concerns. For reference, we have included line numbers relevant to the manuscript without tracked changes.
* * *
**Reviewer 2**

The paper is overall a useful contribution and well written. I hope that my comments below will improve its impact to the larger hydrologic community. I have blended below both higher level comments (approach for have more impact for the international community) and more specific for improving logic. I have no problem with its publication but I am also not impressed with the message it tries to send to the community and what it can accomplish to improve a vision for HOs. Also, its focus on the "UPH" limits its reach and the community that would buy into it, while a more high-level approach of basically understanding complex hydrologic processes to improve modeling and prediction that will allow us to address pressing water related problems… will reach a larger audience and sponsors.

**REPLY**: We would like to thank this reviewer for reviewing our work. We concur that improving hydrological modeling within a hydrological observatory is crucial for a more comprehensive understanding, prediction, and management of water resources. Data collection is the key component to perform reliable modeling simulations of water balance, solute and heat, transport, and soil erosion. The use of sensors with greater density and data interpolation helps capture a diverse range of hydrological processes. It is also crucial to include human impacts on model simulations, such as land-use change, water resources management practices, and the impacts of global warming on the hydrological systems. Nevertheless, the initial step is to formulate a new UPH, as previously discussed at the beginning of Section 3. The new UPH will dictate the HO functioning and model implementation/refinement.

The following sections address Ref.#2 concerns and integrate the relevant suggestions where appropriate. In this response, line numbers are referenced to the manuscript without the use of tracked changes. Any elements that might have caused confusion have been removed, and several points have been clarified.

1. Line 36 -- "Nevertheless we are still a long way from being able to solve the mysteries of hydrologic processes…" – the mysteries of many scientific problems are never completely solved. I would present this differently such as " Yet, solving important water resources problems requires a deep understanding of the complex hydrologic processes which require long records of observations over diverse environments etc.…"

**REPLY**: To avoid repetitions, the first part of the abstract was reformulated in a manner that partially accommodated this suggestion. In lines 33-40 we wrote: "*The Unsolved Problems in Hydrology (UPH) initiative has emphasized the need of establishing networks of multi-decadal hydrological observatories to gain a comprehensive understanding of the complex hydrologic processes occurring in diverse environments. The already existing monitoring infrastructures have provided an enormous amount of*

*hydrometeorological data, facilitating detailed insights into the causality of hydrological processes, the testing of scientific theories and hypotheses, and the development of physical laws governing catchment behavior. Yet, hydrological monitoring programs have often produced limited outcomes because of the intermittent availability of financial resources and the substantial efforts required to operate observatories and conduct comparative studies to advance previous findings.*"

2.Line 46 – "help address UPH about the impact of climate and social systems…" –First, is this the only UPH to address from the whole list of UPH? Second, I would present this need here in a more general setting. Recall that one does not even know the long list of UPH and if this is an international effort it has to be presented from an even larger perspective …
**REPLY**: We agree and, as a result, this part has been reformulated by accommodating this suggestion. In lines 45-48 we report: "*A network of moderately instrumented monitoring sites would provide a broad spatial coverage across the major pedoclimatic regions by supporting cross-site synthesis of the lumped hydrological response (e.g., rainfall-runoff relationship, Budyko analysis) across diverse continental landscapes. However, the moderate instrumentation at each site may hamper an in-depth understanding of complex hydrological processes.*"

3.Fig 1 is ok but again, cross site synthesis is not the key to many problems but depends on the problem to be addressed… as also articulated later in the paper for ocean missions etc.
**REPLY**: We decided to remove this figure and the corresponding text as suggested by the first reviewer. We prefer to give more emphasis to the last part in which we compare two different scenarios.

4.Line 82 – I would strongly suggest that the title of this section is changed to something like "The need for HOs to advance scientific understanding of hydrologic processes " instead of "How to address the UPH" for which probably not everyone agree or might have a different problem not included in that list!
**REPLY**: We agree with this comment and modified the title:
"*How do we advance scientific understanding of hydrological processes?*"

5.Line 89 – "the extent that anthropogenic stressors influence the hydrologic cycle is not yet fully understood…" – I would argue that if we know the stressor then we can address the forward problem of translating it to an outcome or impact, but the challenge is when we do not know what actions will affect what and how, and we need basic understanding to guide decisions and management for guiding the future of water…
**REPLY**: We reformulated this sentence by including the suggested comment. The new sentence in lines 88-89 is:
"*However, the extent to which anthropogenic stressors influence the hydrologic cycle remains unclear, and the effectiveness of adaption actions to guide water resources managers has not been fully evaluated.*"

6.Line 95 – HOS are not always long-term sites
**REPLY**: We removed "long-term" to avoid confusion

7.Lines 98-99 – stretching it by much here. If this is to have an international and broad audience, this has to be seen from a higher level. CZOs, NEON etc had nothing to do with the UPH, as an example… -- check their vision when established
**REPLY**: We understand that the majority of currently operational observatories worldwide are guided by interdisciplinary research goals that extend beyond the scope of UPH. Such observatories are, in fact, defined as terrestrial observatories. In this opinion paper, however, we will limit our discussion only to the hydrological aspects. The proposed hydrological observatories can be part of comprehensive terrestrial observatories, such as eLTER. We integrated the text in lines 209-215 to accommodate this suggestion: "*To address these issues, scientists have proposed initiatives to sustain long-term operation, harmonize, and standardize both hydrometeorological data and eco-hydrological models in HO networks (Zoback 2001; Reid et al., 2010; Kulmala, 2018). In many cases, hydrological observations are now integrated into interdisciplinary research programs within terrestrial observatories, which are scientific facilities designed to observe and study a range of aspects related to the Earth's surface, atmosphere, and interior. Terrestrial observatories collect data on various phenomena, including earthquakes, volcanic activity, weather patterns, climate change, and the movement of tectonic plates. Hydrological observations play a crucial*

*role in the context of terrestrial observatories.*" Indeed, in lines 215-226 we mention some examples of hydrological and terrestrial observatory networks. Hydrological observations play a crucial role in terrestrial observatories.

8.Lines 104 – check history papers for some early observatories of Horton (Beven special IAHS volume)
**REPLY**: We found the following reference:

Beven, K. J. (2006), Streamflow Generation Processes, 431 pp., IAHS Press, Wallingford, U. K.

However, adding historical papers is not necessary as some important very early papers have already been cited.

9.Line 109 – evidence for this exponential growth?
**REPLY**: To avoid confusion, we reformulated this sentence: "*The number of HOs has increased since the 1950s by setting many HOs across the globe*" (line 106)

10.Lines 121-122 – will benefit from some editing
**REPLY**: We reformulated this part: "*The selection of sensors is crucial for the effective collection of hydrometeorological data within a hydrological observatory.*" (line 126-127)

11.Line 124 – we are beyond this and LiDAR can help with determining surface flow paths etc with a lot of developments over the past decade
**REPLY**: We agree with this comment, indeed we mentioned the use of LIDAR snow depth surveys in line 125 of the original manuscript.

12.Lines 151 on – RS observations are not only to upscale or downscale ground observations but to provide data for larger areas extents and different environments, and the limited ground observations play a fundamental role in that
**REPLY**: We agree with this comment and we expanded the original sentence: "*The use of unmanned aerial systems (UAS; e.g. Dugdale et al., 2022; Romano et al. 2023) and satellite platforms (e.g. Durand et al., 2021, De Lannoy et al., 2022) provide valuable supplementary information to ground-based observation in HOs. This information can be used for obtaining data over large, heterogenous areas, and for upscaling or downscaling hydrological variables (e.g., McCabe et al., 2017; Manfreda et al., 2018, 2024; Su et al., 2020).*" (lines 157-161)

13.Line 166 – some discontinuity in arguments and logic here
**REPLY**: The text referred to Fig. 1 (lines 166-169 in the original manuscript) and Fig. 1 were removed by following both reviewers' suggestions

14.Fig 1 is ok but not too telling
**REPLY**: Fig. 1 was removed by following both reviewers' suggestions

15.Line 185 – only SMAP? Precipitation is the most important input to the hydrologic cycle and some reference to GPM, IMERG etc should be given, probably also highlighting the successful international cooperation of NASA, JAXA and ESA…
**REPLY**: This section is not about the use of remote sensing products, but about data assimilation in general. The focus of this paper should be on the HO instrumentation with in-situ sensor technology and an exhaustive listing of the numerous remote sensing products is not within the scope of this paper. In any case we reformulated the entire part (lines 175-189): "*While observations are the backbone of progress in hydrological understanding (Sivapalan and Blöschl, 2017), models are equally vital for hypothesis testing and making predictions of practical relevance (Brooks et al., 2015; Baatz et al., 2018; Bogena et al., 2018; Bechtold et al., 2019; Nearning et al., 2024). However, hydrological models, particularly complex ones, often rely on lumped parameter calibration. This means that model parameters are adjusted based on aggregated (or lumped) fluxes, such as streamflow measurements at the catchment's outlet. While this approach can be effective, it can also lead to limitations. One significant challenge is the assumption that the model's behavior is uniform across the entire catchment. This assumption might not hold true, especially*

*in heterogeneous catchments with varying topography, land use, and soil types. In such cases, relying solely on lumped fluxes can result in suboptimal model performance. An integrated observation approach enables the calibration based on insightful analysis of process complexity through systematic learning from distributed hydrometeorological data given that catchments are complex systems with structured heterogeneity that give rise to non-linear interactions and feedback between the component processes (Vereecken et al., 2015; Li et al., 2022). One way of model-observation integration is the assimilation of observations into hydrological models (Mwangi et al., 2020; Kumar et al., 2022; De Lannoy et al., 2022) to estimate unobserved variables, improve predictions, and calibrate and validate satellite retrieval (Colliander et al., 2021).*"

16. Fig 2 – UPH is everywhere and distractive. This is not the mission here but process understanding in general. The figure says …"Where the UPH addressed?" No or yes, and depending on the answer we follow a path of "refine approach" or "Hydrological understanding" … First, fundamental questions change and a long-term vision from HOs should not be tied to a limited concept of questions not everyone probably has seen or agrees with…
**REPLY**: In accordance to Reviewer#1's suggestion we removed also Fig. 2. We agree that the design of the HO should not be tied solely to the UPH, but to fundamental hydrological processes. Nevertheless, we believe that the appropriate selection of UPH can support the design of HO. The key factors underlying the planning of HOs are:

1. Research objectives: What specific hydrological processes are you interested in?
2. Spatial and temporal scales: What is the desired resolution of your data?
3. Budget constraints: What is the available funding for sensor acquisition and maintenance?
4. Data management capabilities: How will you handle the volume of data generated?
5. Sensor reliability and accuracy: What level of precision is required?
6. Model selection: What kind of eco-hydrological model are you going to use?

We will make this clearer in the revised version.

17. Line 245 – yes! "Formulate scientifically interesting questions …" not follow "prescribed questions…"
**REPLY**: We modified the second section, and we kept in mind all previous suggestions provided by this Reviewer.

18. Fig 3 is ok but not impressive
**REPLY**: We prefer to keep it to help understand the steps of cross-site synthesis

19. Line 276 – It depends on so many other variables so it is hard to throw this statement as a contradiction …
**REPLY**: We wanted to report some examples of cross-site synthesis. Sometimes the site comparisons lead to conflicting hypotheses and theories that certainly depend on many factors (some of them though remain still unknown or unexplored)

20. Line 278 – "observed phenomena" – which phenomena?
**REPLY**: We reformulated this sentence as: "*Two distinct theoretical frameworks have been put forth to explain the above mentioned conflicting results (Ellison et al., 2012).*" (lines 277-278)

21. Line 312- 316 –Yes, these observatories were designed for specific scientific questions not for "UPH" -- resonates much more with the community at large.
**REPLY**: OK

22. Line 317 – In analogy with the above questions, what would be examples of questions to be addressed by these sites?
**REPLY**: Please refer to the reply given to Reviewr#1. We added a new Table by grouping the UPH according to each management scenario in the Appendix.

23.Line 324 – Yes I agree with this. This contradicts the whole framing of the paper focused on "the UPH"! Also, the arguments in Lines 335-on defeat the arguments on the starting point of this paper.
**REPLY**: We tried to follow this suggestion throughout the manuscript. Thank you for pointing it out.

24.Line 374 – Yes but as argued above deep observations in one site can significantly knowledge our knowledge in important problems. Some examples as from the CZOs. So there are some contradicting staments here referring to a "global network" etc. Please check.
**REPLY**: We considered this suggestion to modify some parts of the manuscript

25.Line 393 – 399 -- "We envision a hybrid approach …" Yes, ok but how? This is the question and the end of the paper kind of fails to have a "call to action" and inspire a movement. It is a difficult problem of course but the paper left me at the end with no recommended approach …
**REPLY**: The main target of this opinion paper is to raise critical discussion on the management of HOs. It is beyond the scope of this opinion paper to provide a manifesto on how to plan and run a hypothesized "hybrid" management approach. This process would require a focused report by inviting stakeholders, research institution, governmental actors, etc.

REFERENCES
Bogena, H., Schulz, K., and Vereecken, H.: Towards a network of observatories in terrestrial environmental research, Adv. Geosci., 9, 109–114, https://doi.org/10.5194/adgeo-9-109-2006, 2006.

---

## Author Response (AR2)

**HESS Opinion paper:**
**Towards a common vision for the future of hydrological observatories**

**EDITOR**

We would like to thank the Editor for handling our contribution. The first reviewer provided very positive and extremely constructive comments, which helped us to substantially improve the quality of our manuscript. We have modified some parts of the manuscript according to his suggestions. The second reviewer raised concerns especially about the initial part of the manuscript. Therefore, we have removed the first two figures (which were actually not well received by both reviewers) from the second section and expanded the last part of the manuscript, which deals with the comparison of two hypothetical scenarios for the management of hydrological observatories (HOs). In response to the first reviewer's suggestion, we have added two new figures (Figures R1 and R2 reported in this reply letter) to Section #4 to provide further emphasis on the comparison of the two management scenarios.

We have revised the original manuscript submitted to Hydrology and Earth System Sciences, and we do hope that the new version addresses most of the reviewers' concerns. For reference, we have included line numbers relevant to the manuscript without tracked changes.

**Reviewer 1**

Review by Andrew J. Guswa, Smith College, aguswa@smith.edu

This editorial, which grew out of discussions at the 8th Galileo Conference held in Napoli, Italy in June 2023, raises the question of how funding and resources for hydrological observatories could best be deployed, considering two scenarios: 1) a large number of moderately instrumented sites or 2) a small number of highly instrumented "super sites" (the dynamic-response approach notwithstanding). The question is a provocative and important one. With the current version of the manuscript, I also found myself wishing for more details of the two models, more discussion of the scientific tradeoffs between them, and perhaps considering ideas beyond the two. Additionally, I think it would be interesting and valuable to discuss potential differences in governance structures, data access, and equity issues. To make room for such detail and discussion, I think some of the preamble/motivation could be shortened and one or two of the figures removed/replaced.

**REPLY**: We would like to thank Ref. #1 (A. Guswa) for reviewing our work. In the following sections, we have addressed his concerns and incorporated his suggestions where appropriate. In this response, line numbers refer to the untracked manuscript. We have removed elements that may have caused confusion and clarified several points.

Specifically, perhaps Figure 1 could be expanded into two – one that presents the instruments and their distribution associated with a site for Scenario 1, and a second figure that presents the vision for a super-site associated with Scenario 2. Would they have the same types of instruments, but with different levels of intensity and resolution? Or would there be additional variables and characteristics measured in the super-sites? Providing more concrete detail – even if purely hypothetical – will help the reader better understand the two models and the differences between them.

**REPLY:** Before illustrating the main differences between the two management scenarios, a hydrological observatory is defined as the cyber-physical infrastructure established within a catchment to monitor the hydrological variables and fluxes and to characterize the hydrological behavior of the 3D spatial domain. The catchment is assumed to be the fundamental hydrological unit with well-defined system boundaries. It is from this unit that the impact of anthropogenic perturbations (global warming, land use change, aquifer contamination, etc.) on water resources can be assessed through a long-term data analysis. Given the impracticability of covering the entire catchment area, the hydrological observatory focuses on a cluster of

representative sub-catchments (spatial resolution of hectares), which are representative in terms of land use, geomorphology, topography, and pedology similarities (Bogena et al., 2006). Therefore, the selected sub-catchments are equipped with wireless sensor networks for continuous data collection and subjected to different field campaigns depending on budget constraints.

As described in the original manuscript (Section 4), we have assumed that a fixed community budget has been allocated for the establishment and operation of a hypothetical network of HOs in the European Union (EU). Figure R1 shows an example of a moderately instrumented site belonging to a network of hydrological observatory in Scenario 1. This plan reflects the current situation at most HO networks around the world. S1 offers the clear advantage of widely distributed HOs across each continent, enabling cross-site synthesis of the lumped hydrological response (e.g., rainfall-runoff relationship, Budyko analysis) across diverse continental landscapes.

[Figure]

*Figure R1. Graphical illustration of a hydrological observatory (HO) network in the European Union (EU) in Scenario 1. Each sub-catchment is equipped with basic instrumentation: a weather station, a runoff gauging station, a Cosmic-Ray Neutron Sensor (CRNS) with a wireless sensor network controlling soil profile sensors, a streamflow sensor at the catchment's outlet. Satellite products are available anywhere in the world. The soil profile cross-section illustrates the soil profile sensor unit and the stationary CRNS.*

Figure R2 delineates a hypothetical network of super-sites established along an ideal transect within Europe (Scenario 2). Each super-site should have a high-density network of sampling and monitoring units for soil hydrology research. This infrastructure, as yet unrealized, would facilitate a comprehensive understanding of water dynamics in the groundwater-soil-plant-atmosphere continuum, and of surface and subsurface water circulation at a few sites on each continent. In this case, cross-site synthesis would support the application and refinement of complex hydrological models based on fundamental insights into complex, non-linear processes modulated by scale-dependent feedbacks and multiscale spatiotemporal heterogeneity.

[Figure]

*Figure R2. Graphical illustration of a hydrological observatory (HO) network in the European Union (EU) in Scenario 2. Each sub-catchment established along an ideal transect is equipped with high-density network of sampling and monitoring units for soil hydrology research. Frequent unmanned aerial system (UAS) and aircraft surveys are organized over the experimental area. Satellite products are available anywhere in the world. Frequent campaigns of geophysical (electromagnetic induction, EMI technique) and tracing (stable isotopes in water such as $\delta^2H$ and $\delta^{18}O$) measurements are carried out across the HO. Flow monitoring and water sampling are carried out along the stream. The soil profile cross-section shows the monitoring and sampling activities in the groundwater-soil-plant-atmosphere continuum in a position of the dense point grid (purple circles).*

We thoroughly revised Section 4 by following this suggestion

Then, given those differences, which of the UPH would be more amenable to Scenario 1 versus Scenario 2? The Appendix provides a list of important and intriguing questions – which would be better addressed by which Scenario? The manuscript already calls out differences in representativeness: e.g., Scenario 1 can cover more of "existing gradients of geology, climate, and land-use," whereas Scenario 2 would achieve "high spatial and temporal resolutions." More directly connecting these differences to the important questions would be a valuable enhancement to the paper.

**REPLY**: In our view, all of the UPH mentioned below (and many others) are susceptible to both scenarios, which are not mutually exclusive but rather mutually reinforcing concepts. For example, detailed investigations necessitating unparalleled instrumentation in S2, would facilitate the transfer of acquired knowledge to other biogeographical regions in S1. We aim to clarifying this point in the revised article.

To respond to this reviewer, we have grouped the UPH according to the associated Scenario:

**Questions in Scenario 1**

| | |
|---|---|
| 1 | How might the quantity and quality of measurements be improved in data-poor regions? |
| 2 | Where and how can the sensors be allocated to get full information without wasting excessive effort? |
| 3 | Are measurements taken in the past still valid in the future? How about accuracy/precision change with technological advancements? Do we need to remove all "inaccurate" historical data and keep only "currently accurate" data? Is the assumption of a steady hydrological system valid? Can we simplify the system by linearizing a nonlinear system behavior? |
| 4 | What role(s) do continuous and ephemeral water bodies, including ponds, lakes, rivers, streams, marshes, swamps, and so forth, play in influencing the quantity and quality of water in a catchment? |

**Questions in Scenario 2**

| | |
|---|---|
| 1 | What are the hydrologic laws at the catchment scale, and how do they vary with scale? |
| 2 | How can we use innovative technologies to measure surface and subsurface properties, states, and fluxes at a range of spatial and temporal scales? |
| 3 | How can the assimilation of multi-scale observations into a hydrological model enhance the model's predictive capacity? |
| 4 | How might we obtain large-scale flux measurements and feedback to analyze the water dynamics within and between the various compartments of the groundwater-soil-plant-atmosphere continuum? |
| 5 | How is the water cycle influenced by the other cycles (carbon, nitrogen, etc.)? |
| 6 | How can the dynamics and feedback at groundwater-soil, groundwater-surface water, soil-plant, soil-atmosphere, and plant-atmosphere interfaces be assessed? |
| 7 | How do we include plant physiological aspects in hydrological models? |

**Questions in both scenarios**

| | |
|---|---|
| 1 | What is the impact of preferential flow on catchment-scale water flow dynamics? |
| 2 | How might remote sensing be employed to provide more reliable information on soil moisture, changes in water storage, surface energy balance, and evapotranspiration at suitable spatial and temporal scales (Lettenmaier et al., 2015)? |
| 3 | What are the causes of spatial heterogeneity and homogeneity in runoff, evapotranspiration, subsurface water and material fluxes (carbon and other nutrients, sediments), and in their sensitivity to their controls (e.g., snowfall regime, aridity, reaction coefficients)? |
| 4 | How can hydrological models be adapted to be able to extrapolate changing conditions, including changing vegetation dynamics? |
| 5 | How might we disentangle and reduce model structural/parameter/input uncertainty inherent in hydrological predictions? |
| 6 | Should greater emphasis be placed on uncertainty or causality? |
| 7 | How do vegetation types, distribution, and dynamics shape hydrological processes, particularly in terms of water quality, quantity, and energy fluxes at the catchment scale? |
| 8 | How can we integrate the different spatial and temporal scales of observations, processes, and models? |
| 9 | How can we develop socio-hydrological models by considering anthropogenic disturbances in the ecosystem? |

We highlight the predominant number of UPH valid for both scenarios.

Additionally, I found myself wondering about the associated tradeoffs in governance, equity, site access, and data availability between the two scenarios. Perhaps this is not the place for such discussions; nevertheless, if possible, I think a modest discussion of those issues would be interesting and useful. What processes could/would be put in place to encourage participation from a broad range of stakeholders from the hydrologic community, particularly for Scenario 2?

**REPLY**: Thank you for this valuable comment. In the revised version, Section 4 has been expanded to include a more detailed discussion of the trade-offs between the two scenarios in terms of governance, site access, and equity.

**Scenario 1 (S1):** S1 relies on a combination of centralized and distributed components. Distributed components provide observed data managed by different entities (e.g., universities, research institutions, government agencies, etc.) across geographically spread sites. To ensure data comparability, it is essential to implement standardized protocols for data collection, storage, quality assurance and analysis. This reduces the effort required for cross-site synthesis. In addition, centralized data management facilitates access to data across multiple sites. Moreover, additional central thematic elements can be provided, such as those pertaining to communication and knowledge transfer, or those relevant to modeling applications. The organizational structure can be based on other successfully established or planned distributed continental infrastructures. Notable examples include ICOS (Integrated Carbon Observation System) or eLTER (Integrated European Long-Term Ecosystem, Critical Zone and Socio-ecological Research Infrastructure). Free availability of data and accessibility of the sites should be a fundamental aspect of scenario design.

Collaboration and partnerships between different stakeholders are crucial in S1, which could provide broader opportunities for citizen and stakeholder participation, particularly given the distributed nature of the scenario and the promotion of local initiatives.

**Scenario 2 (S2):** Few super-sites would require a central governing body that would likely be responsible for overseeing all aspects of the super-sites, including instrument deployment and maintenance, as well as data collection and analysis. Such an entity could be a dedicated government agency with a specific mandate or a research consortium with substantial resources.

The establishment of single entity acting as a central authority would facilitate a more streamlined decision-making process regarding instrument upgrades, research focus, and site and data access.

Super-sites equipped with advanced instrumentation could attract highly specialized researchers of different disciplines, resulting in a concentration of knowledge and experience in specific hydrological domains. The implementation of standardized sensors would lead to cost savings and improved efficiency in data collection and processing. In contrast, different hydrological environments may require specialized instrumentation or measurement techniques. A lack of flexibility in standardization may impede the ability to adapt to new research questions or emerging challenges. Examples of standardization efforts include the Global Network of River Observatories (GLORIA; https://www.gloria.ac.at) and the World Meteorological Organization (WMO; https://community.wmo.int/en) guidelines for hydrological stations. By carefully considering these factors and adopting a balanced approach, hydrological observatories can harness the benefits of standardization while maintaining flexibility and adaptability. To ensure equity and encourage greater participation in S2, it is essential to establish a collaborative governance structure that involves a wide range of stakeholders in decision-making processes related to super site operations and data utilization. The governance and site access aspects are well presented in initiatives such as the International Continental Scientific Drilling Program (ICDP), which addresses geodynamic processes, solid Earth geohazards, sustainable geo-resources, and environmental change (https://www.icdp-online.org/about-icdp/entities/). Another relevant example is the Alfred Wegener Institute (AWI), which aims to understand the complex processes in the Earth system and the impact of global warming on the oceans and polar regions (https://www.awi.de/en/). The AWI maintains a network of well-instrumented long-term observatories, comprising both stationary devices and mobile components that are employed for studies related to oceanography, meteorology, and geophysics (https://www.awi.de/en/expedition/observatories.html).

Relatedly, while the authors position their two scenarios as end-members, I could imagine a scenario further in the direction of Scenario 1 (perhaps Scenario 0), in which instrumentation is deployed in a purely opportunistic way, taking advantage of construction projects associated with infrastructure upgrades. For example, one could imagine a policy that stipulates that any time a culvert is rebuilt or rehabilitated (e.g., in response to increasing storm intensity), a suite of monitoring instruments must also be installed. This would significantly reduce the upfront costs associated with hydrologic observation. The locations of such added monitoring would be far from planful, but there might be advantages in the sheer number of sites, and ongoing advances in data handling/storage and machine-learning and data-science tools could facilitate new insights.

**REPLY**: We agree that by leveraging infrastructure upgrades for hydrological monitoring offers great opportunities to significantly reduce initial costs compared to building new monitoring sites from scratch. Nevertheless, the opportunistic approach is applicable to all scenarios.

To make space for those expanded discussions, I think either Figure 2 or Figure 3 could be removed. Those schematics are nice, but perhaps are not necessary for communicating the central ideas of the paper (vs. the Graphical Abstract, which really presents the Scenarios in a compelling way).

**REPLY**: We concur with this comment and decided to remove the first two figures while expanding the discussion in the last part. Here, we have added the new figures presented in the reply letter (R1 and R2 in this reply letter). The new figures illustrate a hypothetical super-site in comparison to a moderate site.

In summary, I appreciate the editorial as a provocation for the hydrologic community to consider and discuss the vision for hydrologic observations. I recommend expanding the comparison of the two scenarios to help the reader better understand the tradeoffs between them.

**REPLY**: We concur with this assessment and will therefore shorten the initial section as much as possible while expanding the discussion of the trade-offs, including the issues of governance, site access, and equity.
* * *
**Reviewer 2**

The paper is overall a useful contribution and well written. I hope that my comments below will improve its impact to the larger hydrologic community. I have blended below both higher level comments (approach for have more impact for the international community) and more specific for improving logic. I have no problem with its publication but I am also not impressed with the message it tries to send to the community and what it can accomplish to improve a vision for HOs. Also, its focus on the "UPH" limits its reach and the community that would buy into it, while a more high-level approach of basically understanding complex hydrologic processes to improve modeling and prediction that will allow us to address pressing water related problems... will reach a larger audience and sponsors.

**REPLY**: We would like to thank this reviewer for reviewing our work. We concur that improving hydrological modeling within a hydrological observatory is crucial for more comprehensive understanding, prediction, and management of water resources. Data collection is the key component to perform reliable modeling simulations of water balance, solute and heat, transport, and soil erosion. The use of sensors with greater density and data interpolation helps capture a diverse range of hydrological processes. It is also critical to include human impacts on model simulations, such as land-use change, water resources management practices, and the impacts of global warming on the hydrological systems. Nevertheless, the initial step is to formulate new UPH, as discussed at the beginning of Section 3. The new UPH will dictate the HO functioning and model implementation/refinement.

The following sections address the concerns of Ref.#2 and incorporate the relevant suggestions where appropriate. In this response, line numbers are referenced to the manuscript without the use of tracked changes. Any elements that may have caused confusion have been removed, and several points have been clarified.

1. Line 36 -- "Nevertheless we are still a long way from being able to solve the mysteries of hydrologic processes…" – the mysteries of many scientific problems are never completely solved. I would present this differently such as " Yet, solving important water resources problems requires a deep understanding of the complex hydrologic processes which require long records of observations over diverse environments etc.…".

**REPLY**: To avoid repetitions, the first part of the abstract was reformulated in a manner that partially accommodated this suggestion. In lines 33-40 we wrote: "*The Unsolved Problems in Hydrology (UPH) initiative has emphasized the need to establish networks of multi-decadal hydrological observatories to gain a deep understanding of the complex hydrologic processes occurring within diverse environmental conditions. The already existing monitoring infrastructures have provided an enormous amount of hydrometeorological data, facilitating detailed insights into the causal mechanisms of hydrological processes, the testing of scientific theories and hypotheses, and the revelation of the physical laws governing catchment behavior. Yet, hydrological monitoring programs have often produced limited outcomes due to the intermittent availability of financial resources and the substantial efforts required to operate observatories and conduct comparative studies to advance previous findings.*".

2. Line 46 – "help address UPH about the impact of climate and social systems…" –First, is this the only UPH to address from the whole list of UPH? Second, I would present this need here in a more general setting. Recall that one does not even know the long list of UPH and if this is an international effort it has to be presented from an even larger perspective …

**REPLY**: We agree and, as a result, this part has been reformulated by accommodating this suggestion. In lines 46-49 we report: "*A network of moderately instrumented monitoring sites would provide a broad spatial coverage across the major pedoclimatic regions by supporting cross-site synthesis of the lumped hydrological response (e.g., rainfall-runoff relationship, Budyko analysis) across diverse continental landscapes. However, the moderate instrumentation at each site may hamper an in-depth understanding of complex hydrological processes.*".

3. Fig 1 is ok but again, cross site synthesis is not the key to many problems but depends on the problem to be addressed… as also articulated later in the paper for ocean missions etc.

**REPLY**: We decided to remove this figure and corresponding text as suggested by the first reviewer. We prefer to give more emphasis to the last part in which we compare two different scenarios

4. Line 82 – I would strongly suggest that the title of this section is changed to something like "The need for HOs to advance scientific understanding of hydrologic processes " instead of "How to address the UPH" for which probably not everyone agree or might have a different problem not included in that list!

**REPLY**: We agree with this comment and modified the title of Section #1 as follows:
"*How do we advance scientific understanding of hydrological processes?*"

5. Line 89 – "the extent that anthropogenic stressors influence the hydrologic cycle is not yet fully understood…" – I would argue that if we know the stressor then we can address the forward problem of translating it to an outcome or impact, but the challenge is when we do not know what actions will affect what and how, and we need basic understanding to guide decisions and management for guiding the future of water…

**REPLY**: We reformulated this sentence by including the suggested comment. The new sentence in lines 90-92 is the following:
"*However, the extent to which anthropogenic stressors influence the hydrologic cycle is not yet fully understood and the effectiveness of adaptation actions to guide the management of water resources has yet to be fully evaluated.*".

6. Line 95 – HOS are not always long-term sites
**REPLY**: We removed "long-term" at line 95-98 to avoid confusion

7.Lines 98-99 – stretching it by much here. If this is to have an international and broad audience, this has to be seen from a higher level. CZOs, NEON etc had nothing to do with the UPH, as an example… -- check their vision when established

**REPLY**: We understand that the majority of observatories currently in operation around the world are guided by interdisciplinary research goals that extend beyond the scope of UPH. Such observatories are indeed defined as terrestrial observatories. In this opinion paper, however, we will limit our discussion only to the hydrological aspects. The proposed hydrological observatories may be part of comprehensive environmental observatories, such as eLTER. We integrated the text in lines 215-221 to accommodate this suggestion: "*To address these issues, scientists have proposed initiatives to sustain long-term operation, harmonize, and standardize both hydrometeorological data and eco-hydrological models in HO networks (Zoback 2001; Reid et al., 2010; Kulmala, 2018). In numerous instances, hydrological observations are now integrated into interdisciplinary research programs in terrestrial observatories which are scientific facilities designed to observe and study various aspects of the Earth's surface, atmosphere, and interior. Terrestrial observatories collect data on a range of phenomena, including earthquakes, volcanic activity, weather patterns, climate change, and the movement of tectonic plates. Hydrological observations play a crucial role in the context of terrestrial observatories.*" Indeed, in lines 221-232 we have mentioned some examples of hydrological and terrestrial observatory networks. Hydrological observations play a crucial role in terrestrial observatories.

8.Lines 104 – check history papers for some early observatories of Horton (Beven special IAHS volume)

**REPLY**: We found the following reference:
Beven, K. J. (2006), Streamflow Generation Processes, 431 pp., IAHS Press, Wallingford, U. K.

However, adding historic papers is not necessary as some important very early papers have already been cited.

9.Line 109 – evidence for this exponential growth?

**REPLY**: To avoid confusion, we reformulated this sentence: "*The number of HOs has increased since the 1950s by establishing many HOs across the globe.*" (line 108-109)

10.Lines 121-122 – will benefit from some editing

**REPLY**: We reformulated this part: "*The selection of sensors is crucial for the effective collection of hydrometeorological data within a hydrological observatory.*" (line 130-131)

11.Line 124 – we are beyond this and LiDAR can help with determining surface flow paths etc with a lot of developments over the past decade

**REPLY**: We agree with this comment, indeed we mentioned about the use of LIDAR snow depth surveys in line 125 of the original manuscript.

12.Lines 151 on – RS observations are not only to upscale or downscale ground observations but to provide data for larger areas extents and different environments, and the limited ground observations play a fundamental role in that

**REPLY**: We agree with this comment and we expanded the original sentence: "*The use of unmanned aerial systems (UAS; e.g. Dugdale et al., 2022; Romano et al. 2023) and satellite platforms (e.g. Durand et al., 2021, De Lannoy et al., 2022) for remote sensing has emerged as a valuable supplementary method to ground-based observation in HOs for gathering information over large heterogeneous areas as well as for upscaling or downscaling hydrological variables (e.g., McCabe et al., 2017; Manfreda et al., 2018, 2024; Su et al., 2020).*" (lines 161-164).

13.Line 166 – some discontinuity in arguments and logic here

**REPLY**: The text referred to Fig. 1 (lines 166-169 in the original manuscript) and Fig. 1 were removed by following both reviewers' suggestions.

14.Fig 1 is ok but not too telling

**REPLY**: Fig. 1 was removed by following both reviewers' suggestions.

15.Line 185 – only SMAP? Precipitation is the most important input to the hydrologic cycle and some reference to GPM, IMERG etc should be given, probably also highlighting the successful international cooperation of NASA, JAXA and ESA…

**REPLY**: This section is not about the use of remote sensing products, but about data assimilation in general. The focus of this paper should be on the HO instrumentation with in-situ sensor technology and an exhaustive listing of the numerous remote sensing products is not within the scope of this paper. In any case we reformulated the entire part (lines 179-193): "*While observations are the backbone of progress in hydrological understanding (Sivapalan and Blöschl, 2017), models are equally essential for hypothesis testing and making predictions of practical relevance (Brooks et al., 2015; Baatz et al., 2018; Bogena et al., 2018; Bechtold et al., 2019; Nearing et al., 2024). However, hydrological models, particularly those of a complex nature, frequently rely on lumped parameter calibration. This means that model parameters are adjusted based on aggregated (or lumped) fluxes, such as streamflow measurements at the outlet of the catchment. Although this approach can be effective, it can also lead to limitations. A significant challenge is the assumption that the model's behavior is uniform across the entire catchment. This assumption might not hold true, especially in heterogeneous catchments with varying topography, land use, and soil types. In such cases, relying exclusively on lumped fluxes may result in suboptimal model performance. An integrated observation approach enables the calibration based on insightful analysis of process complexity through systematic learning from distributed hydrometeorological data given that catchments are complex systems with structured heterogeneity, which give rise to non-linear interactions and feedback between the component processes (Vereecken et al., 2015; Li et al., 2022). One aspect of integration is the assimilation of observations into hydrological models (Mwangi et al., 2020; Kumar et al., 2022; De Lannoy et al., 2022) to estimate unobserved variables, improve predictions, and calibrate and validate satellite retrieval (Colliander et al., 2021).*".

16.Fig 2 – UPH is everywhere and distractive. This is not the mission here but process understanding in general. The figure says …"Where the UPH addressed?" No or yes, and depending on the answer we follow a path of "refine approach" or "Hydrological understanding" … First, fundamental questions change and a long-term vision from HOs should not be tied to a limited concept of questions not everyone probably has seen or agrees with…

**REPLY**: In accordance to Reviewer#1's suggestion we removed also Fig. 2. We agree that the design of the HO should not be tied solely to the UPH, but to fundamental hydrological processes. Nevertheless, we believe that appropriate selection of UPH can support the design of HO. The key factors underlying the planning of HOs are:

1.  Research objectives: What specific hydrological processes are you interested in?
2.  Spatial and temporal scales: What is the desired resolution of your data?
3.  Budget constraints: What is the available funding for sensor acquisition and maintenance?
4.  Data management capabilities: How will you handle the volume of data generated?
5.  Sensor reliability and accuracy: What level of precision is required?
6.  Model selection: What kind of eco-hydrological model are you going to use?

We have made this clearer in the revised version.

17.Line 245 – yes! "Formulate scientifically interesting questions …" not follow "prescribed questions…"

**REPLY**: We have modified the second section, taking into account all of the previous suggestions provided by this Reviewer.

18.Fig 3 is ok but not impressive

**REPLY**: We prefer to keep it to help understand the steps of cross-site synthesis

19.Line 276 – It depends on so many other variables so it is hard to throw this statement as a contradiction …

**REPLY**: We wanted to report some examples of cross-site synthesis. Sometimes the site comparisons lead to conflicting hypotheses and theories that certainly depend on many factors (some of them though remain still unknown or unexplored).

20. Line 278 – "observed phenomena" – which phenomena?

**REPLY**: We reformulated this sentence as: "*Two distinct theoretical frameworks have been put forth to explain the aforementioned conflicting results (Ellison et al., 2012).*" (lines 283-284).

21. Line 312- 316 –Yes, these observatories were designed for specific scientific questions not for "UPH" -- resonates much more with the community at large.

**REPLY**: OK

22. Line 317 – In analogy with the above questions, what would be examples of questions to be addressed by these sites?

**REPLY**: Please refer to the reply given to Reviewer #1. We have added a new table in which we grouped the UPH according to each management scenario in the Appendix.

23. Line 324 – Yes I agree with this. This contradicts the whole framing of the paper focused on "the UPH"! Also, the arguments in Lines 335-on defeat the arguments on the starting point of this paper.

**REPLY**: We tried to follow this suggestion throughout the manuscript. Thank you for pointing it out.

24. Line 374 – Yes but as argued above deep observations in one site can significantly knowledge our knowledge in important problems. Some examples as from the CZOs. So there are some contradicting statements here referring to a "global network" etc. Please check.

**REPLY**: We considered this suggestion to modify some parts in the manuscript

25. Line 393 – 399 -- "We envision a hybrid approach …" Yes, ok but how? This is the question and the end of the paper kind of fails to have a "call to action" and inspire a movement. It is a difficult problem of course but the paper left me at the end with no recommended approach …

**REPLY**: The main objective of this opinion paper is to stimulate a critical discussion on the management of HOs. It is beyond the scope of this opinion paper to provide a manifesto on how to plan and run a hypothetical "hybrid" management approach. Such a process would require a focused report involving research institution, governmental actors, stakeholders, etc.

REFERENCES

Bogena, H., Schulz, K., and Vereecken, H.: Towards a network of observatories in terrestrial environmental research, *Adv. Geosci.*, 9, 109–114, https://doi.org/10.5194/adgeo-9-109-2006, 2006.